JCB Journal of Cell Biology

# The exocyst complex is an essential component of the mammalian constitutive secretory pathway

Conceição Pereira[1]*, Danièle Stalder[1]*, Georgina S.F. Anderson[2], Amber S. Shun-Shion[3], Jack Houghton[1], Robin Antrobus[1], Michael A. Chapman[2], Daniel J. Fazakerley[3], and David C. Gershlick[1]

**Secreted proteins fulfill a vast array of functions, including immunity, signaling, and extracellular matrix remodeling. In the trans-Golgi network, proteins destined for constitutive secretion are sorted into post-Golgi carriers which fuse with the plasma membrane. The molecular machinery involved is poorly understood. Here, we have used kinetic trafficking assays and transient CRISPR-KO to study biosynthetic sorting from the Golgi to the plasma membrane. Depletion of all canonical exocyst subunits causes cargo accumulation in post-Golgi carriers. Exocyst subunits are recruited to and co-localize with carriers. Exocyst abrogation followed by kinetic trafficking assays of soluble cargoes results in intracellular cargo accumulation. Unbiased secretomics reveals impairment of soluble protein secretion after exocyst subunit knockout. Importantly, in specialized cell types, the loss of exocyst prevents constitutive secretion of antibodies in lymphocytes and of leptin in adipocytes. These data identify exocyst as the functional tether of secretory post-Golgi carriers at the plasma membrane and an essential component of the mammalian constitutive secretory pathway.**

## Introduction

The complex process of membrane trafficking is fundamental to cellular organization. Proteins are transported from their site of synthesis in the ER to the Golgi apparatus, where they are sorted to different subcellular localizations, such as the endolysosomal system or directly to the plasma membrane for secretion (Chen et al., 2017; Stalder and Gershlick, 2020). In higher eukaryotes, ~12% of all proteins are secreted from the cell (Kanapin et al., 2003; Uhlén et al., 2019; Thul et al., 2017), where they fulfill a vast array of different functions, including cell signaling, the immune response, and extracellular matrix remodeling (Stalder and Gershlick, 2020).

Soluble secreted proteins are synthesized in the ER. After proper folding, they are trafficked to the Golgi apparatus in COPII carriers, where they are glycosylated (Xu and Ng, 2015; Chen et al., 2017; Patterson et al., 2008; Clermont et al., 1995; Keller et al., 2001). At the trans-Golgi apparatus, soluble secreted proteins are sorted into pleomorphic post-Golgi tubular carriers (Stalder and Gershlick, 2020). These carriers are then trafficked directly to the plasma membrane, where they fuse and their contents are delivered to the extracellular milieu (Polishchuk et al., 2003; Stalder and Gershlick, 2020).

The fusion of intracellular carriers is understood to be a two-step process. Molecular tethers, either long coil-coiled tethers or multisubunit tethering complexes, interact with the carrier prior to the subsequent SNARE-mediated fusion (Lupashin and Sztul, 2005; Whyte and Munro, 2001). The initial "capture" with the tether is therefore essential for correct vesicle targeting and fidelity of cargo delivery.

Long coil-coiled tethers tend to be large (>60 kD) and form a coiled-coil domain structure. Examples of long coil-coiled tethering factors include the golgin family of proteins at the Golgi apparatus, and EEA1 on the endosomes (Lupashin and Sztul, 2005; Murray et al., 2016). They interact with the acceptor compartment on one side and the incoming vesicle on the other, "bringing" the vesicle closer to the target membrane (Murray et al., 2016).

The second class of membrane tether are multisubunit tethering complexes, which include Golgi-associated retrograde protein (Pérez-Victoria et al., 2009) complex on the trans-Golgi network, the conserved oligomeric Golgi (Smith and Lupashin, 2008) complex on the medial-Golgi, and the homotypic fusion and protein sorting (Spang, 2016) complex on the lysosomal-endosomal system. These tend to be large multisubunit assemblies and are sometimes, but not exclusively, complexes associated with tethering containing helical rods (CATCHR), which, when assembled, form helical bundles arranged in tandem through a coiled-coil region at the N-terminus (Chou et al., 2016). Multisubunit tethering complexes have also been found

---

[1]Cambridge Institute for Medical Research, University of Cambridge, Cambridge, UK; [2]MRC Toxicology Unit, University of Cambridge, Cambridge, UK; [3]Metabolic Research Laboratory, Wellcome-Medical Research Council Institute of Metabolic Science, University of Cambridge, Cambridge, UK.

*C. Pereira and D. Stalder contributed equally to this paper. Correspondence to David C. Gershlick: dg553@cam.ac.uk.

to be important for proper SNARE assembly in addition to vesicle catching (Pérez-Victoria et al., 2009).

On the plasma membrane, two molecular tethers have been identified. The long coil-coiled protein ELKS (also: ERC, RAB6IP2, or CAST) localizes to patches on the plasma membrane termed "fusion hotspots" due to the higher frequency of vesicle fusion events at these sites (Deguchi-Tawarada et al., 2004; Monier et al., 2002; Nakata et al., 1999; Wang et al., 2002; Fourriere et al., 2019). ELKS was identified as an interactor and probable effector of all three RAB6 isoforms (RAB6A, A′, and B; Monier et al., 2002). ELKS is implicated in secretion of neuropeptide Y in RAB6, MICAL3, and RAB8 positive carriers (Grigoriev et al., 2007, 2011), and synaptic vesicle tethering to the plasma membrane in a neuronal cell model (Nyitrai et al., 2020).

The second tether associated with the plasma membrane is the CATCHR protein complex exocyst (Wu and Guo, 2015). Exocyst is an octamer composed of EXOC1-8 and was originally identified in yeast as important for secretion based on its localization to the plasma membrane and the Sec phenotype (Stalder and Gershlick, 2020). In mammalian cells, exocyst components localize to the Golgi and plasma membrane as well as at vesicle fusion points (Yeaman et al., 2001; Ahmed et al., 2018; Heider and Munson, 2012). Although exocyst is essential for endosomal recycling to the plasma membrane (Grindstaff et al., 1998; Lipschutz et al., 2000; Langevin et al., 2005; Yeaman et al., 2004; Andersen and Yeaman, 2010; Wu and Guo, 2015; Heider and Munson, 2012), ciliogenesis (Rogers et al., 2004; Zuo et al., 2009; Feng et al., 2012), autophagy (Bodemann et al., 2011; Sáez et al., 2019), innate immunity (Chien et al., 2006; Sáez et al., 2019), and cytokenesis (Chen et al., 2006; Fielding et al., 2005; Neto et al., 2013), the role of exocyst in constitutive protein secretion remains unclear. Inhibition of exocyst with antibodies does not affect delivery of tsVSV-G, a marker of the secretory pathway, to the plasma membrane (Yeaman et al., 2001; Grigoriev et al., 2007); however, this could be due to ineffective inhibition by antibodies as the epitope may not be exposed under certain exocyst structural conformations (Inamdar et al., 2016). Conversely, some evidence indicates that exocyst is important for biosynthetic sorting to the plasma membrane and depletion of EXOC7 decreases tsVSV-G delivery to the plasma membrane (Liu et al., 2007). It is therefore of interest to examine whether exocyst has a direct role in biosynthetic membrane protein sorting in mammalian cells. Moreover, it is not known if exocyst is necessary or important for soluble protein secretion in mammalian cells (Stalder and Gershlick, 2020; Wu and Guo, 2015).

To investigate the functional machinery in protein secretion, we developed a quantitative trafficking assay to study cell-surface delivery from the Golgi apparatus using the retention using selective hooks (RUSH) system. By designing a synthetic type-1 membrane protein based on LAMP1, we can directly observe post-Golgi carriers that co-localize with previously characterized markers and fuse with the plasma membrane. We have used a transient CRISPR-knockout (KO) system to determine that the exocyst complex is essential for the arrival of these carriers to the plasma membrane. We observe exocyst subunits localizing to the post-Golgi carriers on fusion hotspots on the plasma membrane. Kinetic trafficking assays on a set of soluble secreted proteins reveals a broad dependence on exocyst for protein secretion. We performed unbiased proteomics in an endogenous context to exocyst-KO cells and have demonstrated that the exocyst complex is responsible for the majority if not all soluble protein secretion. In addition, we show that important specialized secretory cells require exocyst for the efficient secretion of both antibodies and hormones. We, therefore, define exocyst as the molecular tether for constitutive protein secretion of soluble proteins and as an essential component of the mammalian secretory pathway.

## Results

### Generation of quantitative kinetic cell-surface trafficking assay

In order to study post-Golgi carriers, we have generated a synthetic protein that allows monitoring of protein delivery to the plasma membrane with proper spatiotemporal kinetics. The single-pass type-1 integral membrane protein LAMP1 is localized to the lysosome at the steady-state level. After synthesis in the ER, LAMP1 traffics via the Golgi apparatus directly to the plasma membrane (Chen et al., 2017) where it is endocytosed, in clathrin-coated vesicles, to be delivered to the endolysosomal system and finally to the lysosome (Chen et al., 2017). Mutations in, or deletion of, the endocytic trafficking motif causes LAMP1 to accumulate on the plasma membrane after exit from the Golgi apparatus in post-Golgi tubular carriers (Chen et al., 2017). To monitor the kinetics of trafficking, we used the RUSH system (Boncompain et al., 2012). LAMP1ΔYQTI was genetically fused to a streptavidin-binding peptide (SBP) and a fluorescent protein (GFP) and coexpressed with streptavidin fused to the ER-retrieval signal KDEL (Munro and Pelham, 1987) in a stable cell line (Fig. 1 A).

After biotin addition, the LAMP1ΔYQTI-RUSH (referred to from here as LAMP1Δ-RUSH) can be observed trafficking with appropriate kinetics (Fig. 1 B and Video 1) as previously observed (Chen et al., 2017). Using lattice-SIM (Structured Illumination Microscopy) live-cell imaging, carriers can be observed budding from the Golgi apparatus (Fig. 1 C and Video 2), trafficking along microtubules (Fig. 1 D and Video 3), and using Total Internal Reflection Fluorescence (TIRF) microscopy, we can observe them fuse with the plasma membrane (Fig. 1 E and Video 4). The LAMP1Δ-RUSH is progressively glycosylated after the addition of biotin (Fig. S1), indicating that the protein traffics through the secretory pathway and is processed in the same manner as an endogenous cargo.

To quantitatively study this secretory route, we have developed a flow cytometry–based quantitative cell-surface protein delivery assay (Fig. 1 F). Cells stably expressing LAMP1Δ-RUSH were incubated with biotin across a time series of 1 h, the previously established time to achieve a steady-state level (Chen et al., 2017). By labeling intact cells with anti-GFP nanobody fused to a fluorescent mCherry (Buser et al., 2018) and quantitatively measuring both mCherry and GFP levels via flow cytometry, we can detect and quantify LAMP1Δ-RUSH delivery to

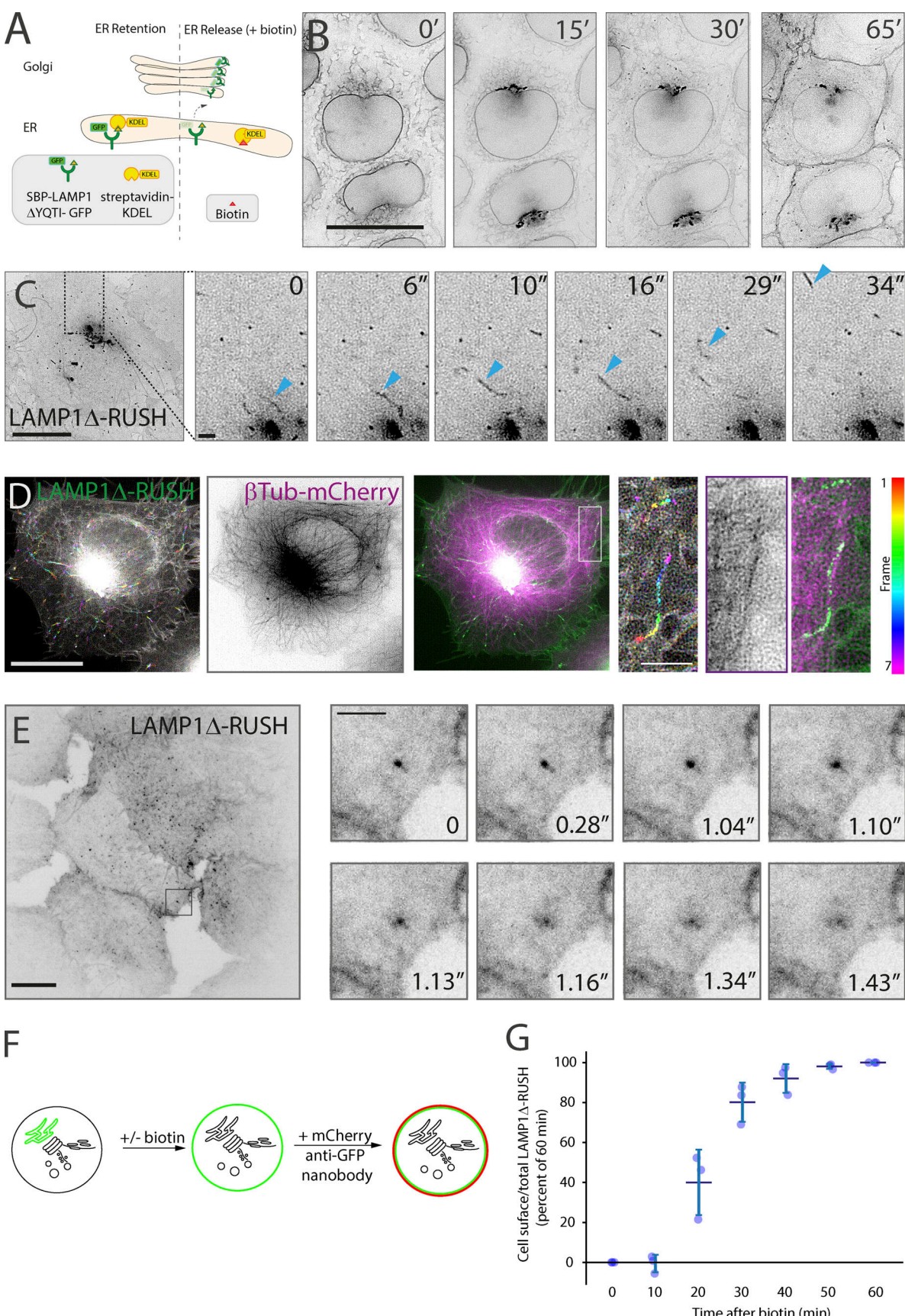

Figure 1. **The biosynthetic LAMP1Δ-RUSH reporter system. (A)** Schematic representation of the RUSH system. By co-expressing the ER hook streptavidin-KDEL with a reporter fused to SBP-GFP, the reporter can accumulate in the ER through the interaction between streptavidin and SBP. Addition of biotin allows

for release of the reporter, which then traffics en masse through the secretory pathway. **(B)** Kinetic analysis of current RUSH cell line based on previous work. The type-1 membrane spanning RUSH reporter LAMP1Δ-GFP is used to monitor transport through the secretory system. Upon the addition of biotin, LAMP1Δ-RUSH traffics from the ER (0′), to the Golgi apparatus (15′), and then directly to the plasma membrane (30′–65′). From Video 1. Scale bar: 10 μm. **(C)** Lattice-SIM imaging allows the observation of LAMP1Δ-RUSH leaving the Golgi apparatus in tubular carriers (indicated by blue arrowheads) 35 min after biotin addition. In the example image (single micrograph from the time series), the cytosol can be seen full of these tubular structures. From Video 2. Scale bar: 10 μm; insert: 2 μm. **(D)** Time color-coded max projection of LAMP1Δ-RUSH carriers (represented by the RGB color bar 0–6s) shows their trajectory over time along the microtubular network (shown as a max projection). The insert shows a close-up example. From Video 3. Scale bar: 10 μm; insert: 2 μm. **(E)** Plasma membrane TIRF plane showing LAMP1Δ-RUSH post-Golgi tubule fusion. Tubules can be seen as bright spots as they approach and fuse, after which the cargo laterally diffuses on the plasma membrane. From Video 4. Scale bar: 10 μm; insert: 2 μm. **(F)** Schematic representation of the RUSH plus cell-surface staining protocol developed for flow cytometry analysis. **(G)** Quantitative cell-surface assay showing time course arrival of LAMP1Δ-RUSH to the plasma membrane after biotin addition. LAMP1Δ-RUSH cell line with a lumenal/extracellular GFP fusion was incubated with biotin for indicated times. Cells were subsequently labeled with a GFP binding nanobody fused to mCherry and underwent single-cell flow cytometry analysis. A minimum of 30,000 cells were analyzed for each biological repeat. The mean of the individual mCherry/GFP cell ratios were calculated for each experiment and are plotted on the graph (blue dots). Error bar = SD of at least three independent experimental repeats.

the plasma membrane. The ratio of the cell surface (mCherry) to total cargo available (GFP) then allows for per-cell quantification of protein cargo arrival at the plasma membrane (Fig. 1 G). In line with previous data, the cargo reaches a steady state ~1 h after release from the ER with biotin. At 35 min, the assay has a high signal-to-noise but is also sensitive to kinetic changes in trafficking, and this time point was selected for future assays. In summary, the use of the RUSH system with a LAMP1ΔYQTI cargo allows for observation of post-Golgi tubular carriers and their fusion with the plasma membrane with appropriate trafficking kinetics and a quantitative readout.

## RAB6A, ARHGEF10, and RAB8A associate with post-Golgi tubular carriers
To validate that the post-Golgi carriers observed using LAMP1Δ-RUSH are representative of secretory carriers, we tested if known markers of these carriers co-localize with LAMP1Δ-RUSH. The small G protein RAB6A/A′ has been observed associated with secretory vesicles that fuse directly with the plasma membrane and has an important role in the fission of the vesicles at the Golgi apparatus and their transport toward the cell surface (Grigoriev et al., 2007; Miserey-Lenkei et al., 2010). Lattice-SIM live-cell imaging showed that overexpressed HALO-RAB6A is associated with tubular carriers emerging from the Golgi apparatus, detaching, and traveling toward the plasma membrane (Fig. 2 A and Video 5). Further, overexpressed HALO-RAB6A co-localized with LAMP1Δ-RUSH at the Golgi apparatus and on these carriers (Fig. 2 B and Video 6), with quantification of carrier co-localization revealing more than 80% of post-Golgi LAMP1 positive carriers have detectable HALO-RAB6A (Fig. 2 C). Another small G protein, RAB8A, associates with exocytotic vesicles in a RAB6-dependent manner through the recruitment of the exchange factor ARHGEF10 (Grigoriev et al., 2011; Shibata et al., 2016). Live-cell imaging demonstrated that both HALO-ARHGEF10 and HALO-RAB8A co-localize with LAMP1Δ-RUSH (Fig. 2, D and F; and Video 7 and Video 8), with significant detectable accumulation on post-Golgi carriers (Fig. 2, E and G) recapitulating previous evidence of a Rab cascade. In conclusion, the post-Golgi carriers observable using LAMP1Δ-RUSH co-localize with a number of markers for secretory carriers.

To demonstrate a functional relationship between RAB6 and LAMP1Δ-RUSH carriers, we transiently abrogated RAB6A using CRISPR-Cas9. Loss of RAB6A drastically and significantly decreased

LAMP1Δ-RUSH trafficking to the plasma membrane (Fig. 2 H). Unlike the inactive form of RAB6A (HALO-RAB6A [TN]), overexpression of Rab6A WT (HALO-RAB6A) as well as the dominant active form (HALO-RAB6A [QL]) rescued the defect in cell-surface delivery of LAMP1Δ-RUSH (Fig. 2 H). RAB6A-KO caused accumulation of LAMP1Δ-RUSH in the Golgi apparatus, which was rescued by overexpressing HALO-RAB6A (Fig. 2, I and J), showing that RAB6A is required for the generation of LAMP1Δ-RUSH post-Golgi carriers. Together, these results show that markers of secretory carriers co-localize with LAMP1Δ-RUSH carriers and are necessary for their trafficking from the Golgi, thus demonstrating the LAMP1Δ-RUSH carriers represent the secretory pathway to the plasma membrane, providing a tractable experimental system to study the secretory pathway.

## Exocyst components associate with fusion hotspots and are essential for carrier delivery to the plasma membrane
Having established a quantitative system for studying the secretory pathway, we utilized it to study the role of plasma membrane tethers in secretory carrier fusion to the cell surface. The long coil-coiled membrane tether ELKS has been previously associated with the fusion of carriers at the plasma membrane (Fourriere et al., 2019). Consistent with these studies, ELKS co-localizes on post-Golgi carrier fusion hotspots by TIRF microscopy (Fig. S2 A and Video 9). Transient or stable KO of ELKS or double KO of ELKS and its characterized cofactors or homologs by CRISPR-Cas9 was performed and validated by immunoblot or quantitative PCR (qPCR; Fig. S2 C and Fig. S3 F). No significant decrease in cell-surface delivery of LAMP1Δ-RUSH to the cell surface could be detected (Fig. S2, B–E).

Based on previous evidence in yeast model systems we hypothesized that the octameric protein complex exocyst is essential for the fusion of post-Golgi carriers with the plasma membrane (Bowser and Novick, 1991; Bowser et al., 1992; TerBush and Novick, 1995). The exocyst is considered an essential component of eukaryotic cells (Novick et al., 1980; Wang et al., 2015), so to avoid lethality and clonal selection artifacts we performed independent transient CRISPR-Cas9 KO of all canonical exocyst components. Using this approach, loss of any canonical significantly decreased cell-surface delivery of LAMP1Δ-RUSH (Fig. 3 A). Rescue with overexpression of EXO-HALO fusions demonstrated that the defect in cell-surface delivery of

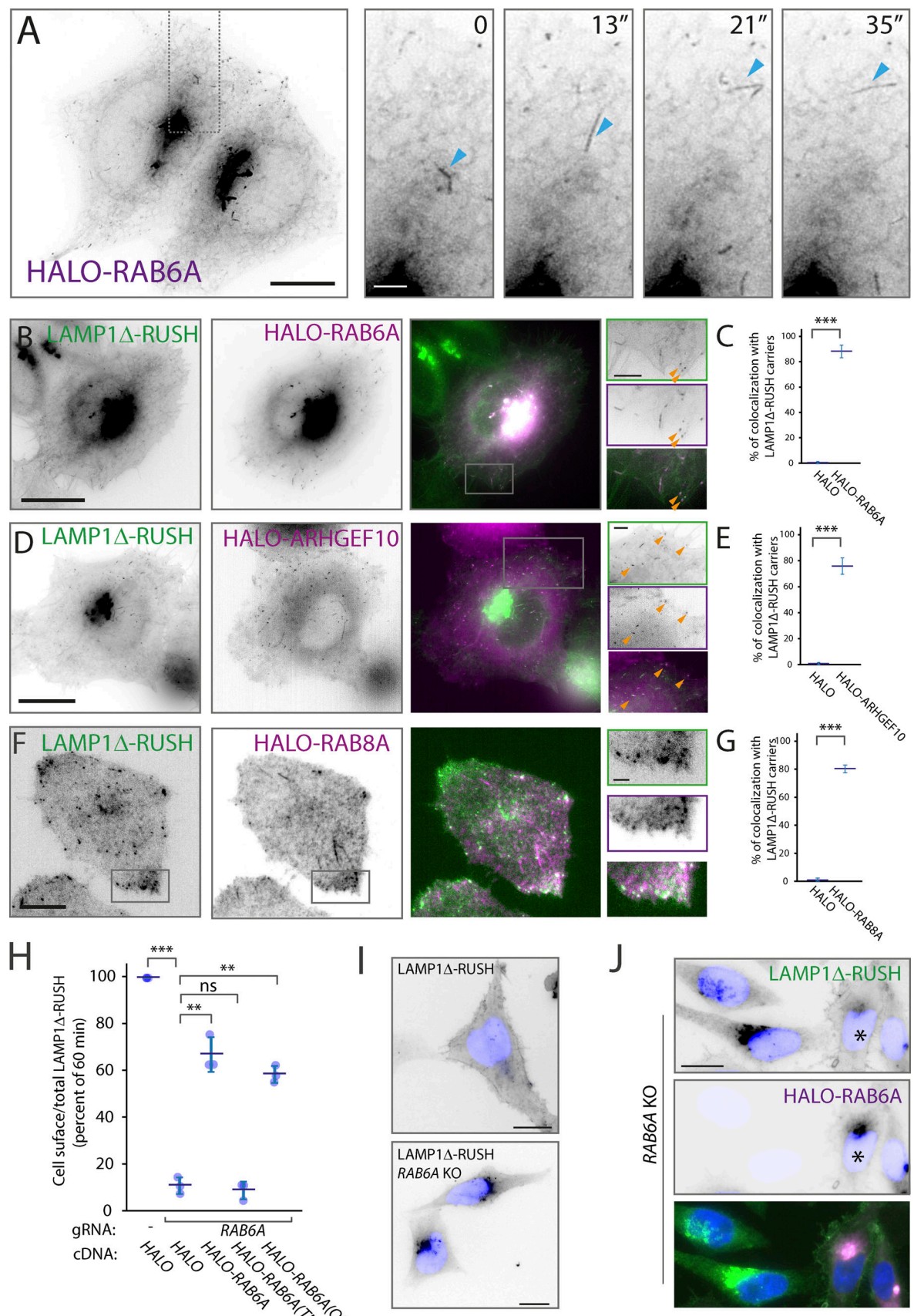

Figure 2. **RAB6A, ARHGEF10, and RAB8A co-localize with LAMP1Δ-RUSH post-Golgi carriers. (A)** Heterologous expression of *HALO-RAB6A* in WT HeLa cells shows RAB6A present at the Golgi and in tubular structures that bud off and travel toward the plasma membrane (single micrograph from a Lattice-SIM

time series), as evidenced by blue arrowheads. From Video 5. Scale bar: 10 µm; insert: 2 µm. **(B)** Lattice-SIM imaging showing HALO-RAB6A (magenta) co-localizing with LAMP1Δ-RUSH tubules (green) leaving the Golgi and moving toward the plasma membrane (~35 min after biotin addition). Orange arrowheads indicate co-localizing structures. From Video 6. Scale bar: 10 µm; insert: 2 µm. **(C)** Percent of LAMP1Δ-RUSH carriers positive for HALO-RAB6A, untagged HALO as a control, three biological repeats (total carriers quantified = 359), statistical analysis = two-tailed *t* test. **(D)** Lattice-SIM imaging showing co-localization of HALO-ARHGEF10 (magenta) with LAMP1Δ-RUSH carriers (green) traveling toward the plasma membrane (~35 min after biotin addition). Orange arrowheads indicate co-localizing structures. From Video 7. Scale bar: 10 µm; insert: 2 µm. **(E)** Percent of LAMP1Δ-RUSH carriers positive for HALO-ARHGEF10, untagged HALO as a control, three biological repeats (total carriers quantified = 389), statistical analysis = two-tailed *t* test. **(F)** Plasma membrane TIRF plane showing HALO-RAB8A (magenta) co-localizing with LAMP1Δ-RUSH carriers (green) near their fusion site at the plasma membrane. The color white denotes co-localizing structures. From Video 8. Scale bar: 10 µm; insert: 2 µm. **(G)** Percent of LAMP1Δ-RUSH carriers positive for HALO-RAB8A, untagged HALO as a control, three biological repeats (total carriers quantified within 3 µm of plasma membrane edge = 285), statistical analysis = two-tailed *t* test. **(H)** Transient RAB6A KO dramatically reduces LAMP1Δ-RUSH at the plasma membrane in a quantitative cell-surface assay carried out 35 min after biotin addition. Note that expression of *HALO-RAB6A* WT (rescue), and constitutively active *RAB6A* (QL), is able to restore plasma membrane expression but not the constitutively inactive form (TN). **(I)** Widefield imaging of LAMP1Δ-RUSH WT and *RAB6A* transient KO, 1 h after biotin addition. Image shows the LAMP1Δ reporter unable to leave the Golgi in the *RAB6A* KO cells. Scale bar: 10 µm. **(J)** Widefield imaging of LAMP1Δ-RUSH reporter (green) in RAB6A KO cells expressing *HALO-RAB6A* (*rescue—magenta). Upon *RAB6A* heterologous expression, the reporter is at the plasma membrane 1 h after biotin addition. Scale bar: 10 µm. Nucleus stain = DAPI. Error bar = SD of at least three independent experimental repeats. Student's *t* test was performed on data in C, E, and G, and Tukey's multiple comparisons test (HSD, FWER = 0.05) was performed on data in H. **P ≤ 0.01; ***P ≤ 0.001.

LAMP1Δ-RUSH is not due to off-target effects or generalized cell lethality (Fig. 3 A). Interestingly, EXOC6-HALO demonstrated poor recovery (Fig. S3 D) in comparison to N-terminally tagged EXOC6 (Fig. 3 A), and overexpression resulted in a moderate but significant decrease in cell-surface arrival of LAMP1Δ-RUSH (Fig. S3 E), compared to N-terminally tagged HALO-EXOC6, consistent with previous reports that C-terminally tagged EXOC6 is experimentally challenging (Ahmed et al., 2018).

Imaging *exocyst*-KO from all subunits of the canonical octameric complex demonstrated an accumulation of post-Golgi carriers at the cell tips (Fig. 3 B), indicating that the point of exocyst involvement in the secretory pathway is after carriers have budded from the Golgi apparatus and prior to fusion with the plasma membrane. Interestingly, double KO of *ELKS* and *EXOC1* resulted in a significant increase in the phenotype (Fig. S2 F), highlighting that ELKS contributes to this pathway. Together these data demonstrate that the exocyst complex is of fundamental importance for cell-surface delivery of LAMP1Δ-RUSH carriers.

**Exocyst is directly recruited to post-Golgi carriers and multiple exocyst-associated proteins are essential for carrier delivery**
To ensure that the effect of *exocyst*-KO on post-Golgi carrier fusion is direct, we tested localization of exocyst components. By tagging EXOC3 with a HALO Tag and expressing it at low levels in our LAMP1Δ-RUSH cell line under *EXOC3*-KO conditions, we observed EXOC3 at fusion punctae co-localizing with LAMP1Δ-RUSH (Fig. 4, A and B, and Video 10). We also observed EXOC1 and EXOC6 co-localizing with post-Golgi carriers prior to fusion with the plasma membrane (Fig. 4, C and D; Fig. S3, A and B; and Video 11 and Video 12). To test the localization of the exocyst subunits to the carriers, we developed an unbiased biochemical approach to localize proteins of interest to post-Golgi carriers. We generated a second LAMP1Δ-RUSH system with the GFP on the cytosolic side of the membrane (Fig. 4 E). This C-terminal fusion had comparable kinetics to the lumenal/extracellular tagged variant (Fig. S4). 35 min after the addition of biotin, there is an accumulation of post-Golgi carriers in the cytosol. We mechanically lysed the cells and immuno-isolated the carriers

using GFPtrap beads, an approach we term "carrierIP." Expression of the control HALO resulted in no enrichment after carrierIP, and all core exocyst components tested (EXOC1, EXOC2, EXOC3, EXOC4, EXOC5, EXOC6, EXOC7, and EXOC8) as well as the positive control RAB6A were enriched on post-Golgi carriers (Fig. 4 F). These data demonstrate that exocyst is directly recruited to the post-Golgi carriers.

Biochemical studies have identified that the phosphatidylinositol-5 kinase (PIP5K) is important for the recruitment of exocyst to the membrane by modifying phosphatidylinositol to phosphatidylinositol-5 phosphate (Maib and Murray, 2021 *Preprint*). There are three PIP5K1 homologs in mammalian cells, PIP5K1A, PIP5K1B, and PIP5K1C. KO of either isoform individually had no detectable effect on cell-surface delivery of LAMP1Δ-RUSH (Fig. 4 G). Triple-transient KO of all three isoforms, however, lead to a decrease of around 60% (Fig. 4 G), comparable to the loss of exocyst (Fig. 3, B–D). Imaging the accumulation of intracellular LAMP1Δ-RUSH in *PIP5K1A/B/C*-KO cells reveals cargo accumulating in post-Golgi carriers prior to fusion at the cell surface consistent with previous observations on exocyst recruitment (Fig. 4 H). Together these data directly tie the machinery known to be important for exocyst recruitment to post-Golgi carriers to the plasma membrane.

**Exocyst is essential for the secretion of multiple soluble cargoes**
There are multiple soluble constitutive cargoes associated with post-Golgi carriers. It is not known whether these cargoes undertake different trafficking routes to the plasma membrane. To find which of these potential routes are dependent on exocyst, we developed a novel RUSH construct in a PiggyBac transposon system backbone (Fig. 5 A) into which we sub-cloned different soluble secreted cargoes. These cargoes included PAUF (CARTS [Wakana et al., 2012]), CAB45 (sphingomyelin carriers [Pakdel and von Blume, 2018]), Collagen X (COLX, RAB6 positive carriers [Fourriere et al., 2019]), NUCB1 (a soluble secreted protein), and signal peptide-HALO (synthetic soluble cargo). We generated stable cell lines from these vectors which resulted in all cargoes being retained in the ER at steady state (Fig. S5) and

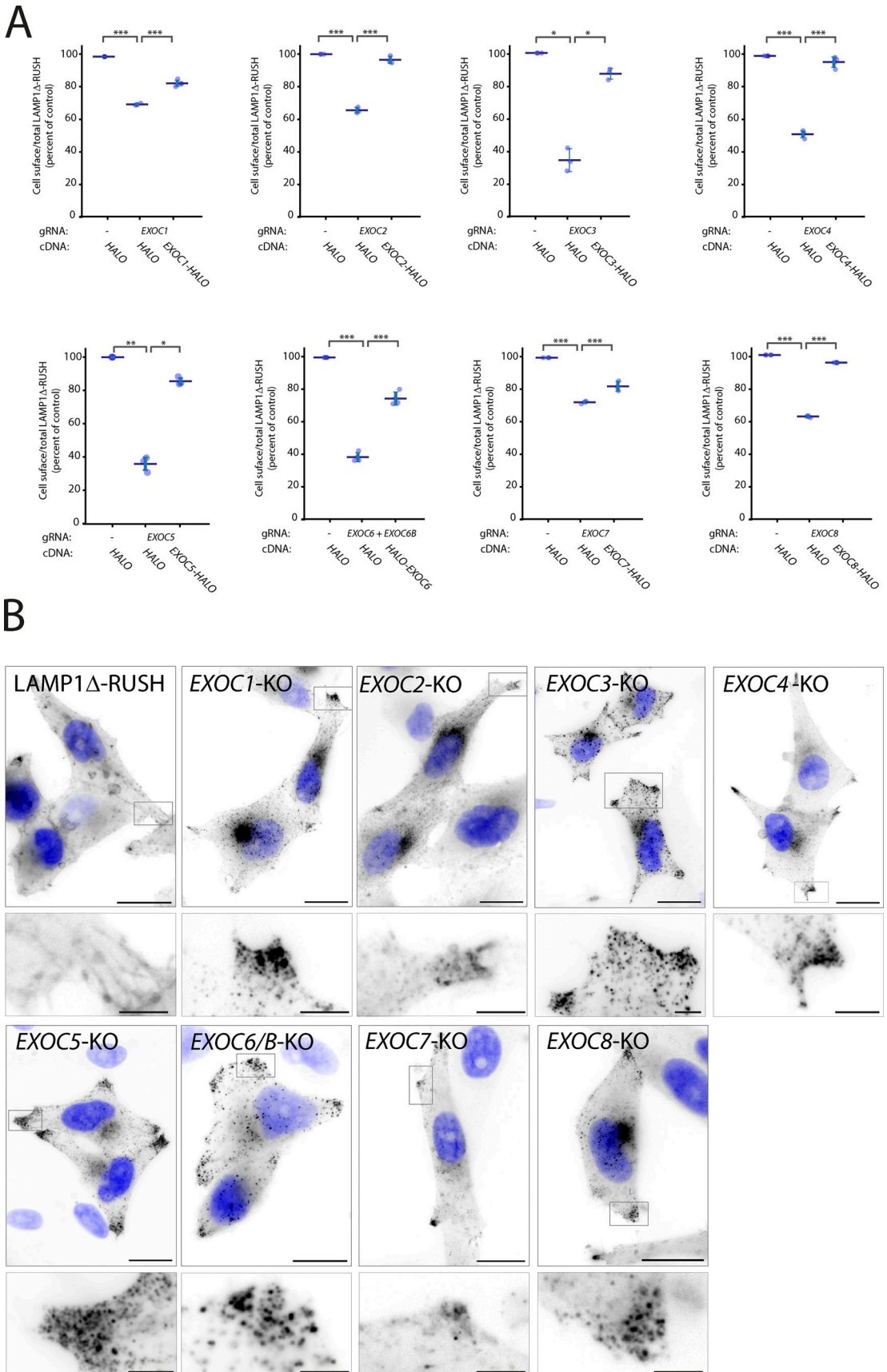

Figure 3. **The exocyst complex subunits are essential for plasma membrane delivery. (A)** Cell-surface ratio quantification (flow cytometry) assay on LAMP1Δ-RUSH at the plasma membrane after individual exocyst subunit KO and recovery with cDNA 35 min after biotin exposure. **(B)** Widefield imaging of

LAMP1Δ-RUSH reporter in exocyst KO cells 1 h after biotin addition. When compared to WT, KO cells show substantial accumulation of post-Golgi LAMP1Δ-RUSH carriers. Scale bar: 20 µm; insert: 4 µm. Nucleus stain = DAPI. Error bar = SD of at least three independent experimental repeats. Tukey's multiple comparisons test (HSD, FWER = 0.05) was performed on data in A. *P ≤ 0.05; **P ≤ 0.01; ***P ≤ 0.001.

subsequently performed transient CRISPR-Cas9 KO of exocyst component *EXOC3*, as before (Fig. 5 B). We then induced RUSH and used an unbiased flow cytometry assay to quantitatively monitor the loss of these soluble cargoes. Intracellular cargo abundance was detected by flow cytometry 12 h after the addition of biotin. Cargo abundance was compared to a non-biotin condition to calculate the amount of cargo secreted by the cells. By incubating the cells with HALO ligand prior to the addition of biotin, we can remove the effect of nascently synthesized cargo in the assay. All cargoes demonstrated a loss of cargo after RUSH, indicating that all proteins were secreted from the cell, which was significantly decreased with the deletion of exocyst (Fig. 5 C).

To evaluate where soluble secreted cargo was accumulating in exocyst abrogated conditions, we performed live cell lattice-SIM imaging. In control cells, no cargo could be seen accumulating intracellularly, supporting the quantitative flow cytometry assay. In *EXOC3*-KO cells, in all conditions, cargoes can be observed accumulating in the cell tips, a phenocopy of the LAMP1ΔYQTI phenotype (Fig. 5 D). Exocyst is therefore essential for the cell-surface delivery of all tested soluble, secreted plasma membrane cargoes.

### Exocyst is essential for secretion of a broad array of soluble secreted proteins

To this point, all experiments have been performed in cells overexpressing proteins in the RUSH system. Thus, the cells are not only overexpressing test cargoes but, in addition, there is a wave of cargo sorting through the cell. To test whether endogenous soluble protein secretion is affected in the absence of exocyst, we performed secretomics on HeLa cells after transient CRISPR-Cas9 KO of *EXOC3*. After filtering the datasets bioinformatically for secreted proteins (i.e., containing a signal peptide, no transmembrane domain, and no glycosylphosphatidylinositol [GPI] anchor), we see a drastic decrease in the secretory profile of cells depleted of exocyst (Fig. 6 A). 51 proteins are significantly (P < 0.05) secreted in the WT cells as compared to the KO cells (Table S2). This included all previously tested proteins that are expressed in HeLa cells (Itzhak et al., 2016; NUCB1, CAB45, and various members of the collagen family). To ensure that this was not due to expression differences in the WT cells compared to the KO cells, we selected several of these (TIMP2, CST3, and PSAP) with commercially available antibodies, to validate these results. As expected, there was an accumulation of these soluble endogenous proteins in *EXOC3*-KO cells. Thus, exocyst is widely essential for the soluble protein constitutive secretory pathway in HeLa cells.

### Exocyst is essential for secretion in professional secretory cells

To test whether exocyst is essential for constitutive secretion in other mammalian cells, we tested this in highly specialized secretory cell models. Adipocytes have a well characterized

regulated exocytosis pathway that is exocyst-dependent (Inoue et al., 2003), as well as a constitutive secretory pathway that is essential for the secretion of leptin, a hormone involved in the regulation of fat stores, the secretion of which has been associated with exocyst (Kuramoto et al., 2021). To test whether secretion of leptin is dependent on exocyst, we abrogated expression of exocyst subunits *Exoc1* and *Exoc3* using siRNA in 3T3-L1 mouse adipocytes (Fig. 6, C and D). We assayed the culture media using ELISA and observed a significant decrease in both leptin secretion in both *Exoc1* and *Exoc3* knockdown conditions (Fig. 6, E and F).

Antibodies are constitutively secreted by blood lymphocytes. To test whether exocyst is necessary for the secretion of antibodies in these cells, we used transient CRISPR-Cas9 to KO *EXOC3* in a clonal myeloma cell line that secretes complete IgG (both heavy and light antibody chains). We incubated the cells for 24 h in fresh media and observed a significant decrease in antibody secretion by immunoblot (Fig. 6, G–I). Together, these data demonstrate exocyst is essential for constitutive protein secretion across multiple different secretory cell types in mammalian cells.

## Discussion

In this study, we have identified the exocyst complex as an essential component of the mammalian secretory pathway. Exocyst localizes to post-Golgi carriers and loss of exocyst prevents delivery of cell-surface carriers, causing them to accumulate in cell tips resulting in a global loss of secretion. In addition, in professional secretory cells such as adipocytes and lymphoma cells, secretion of key proteins such as adipokines and antibodies is heavily reduced upon loss of exocyst.

The role of exocyst as the secretory complex in *Saccharomyces cerevisiae* is well established, due to its original identification in the *Sec* genetic screen (Novick et al., 1980). Exocyst has been previously implicated in biosynthetic sorting in mammalian cells using *ts*VSV-G (Yeaman et al., 2001) which co-localizes with exocyst in the Golgi stacks; however, exocyst inhibition with antibodies did not affect VSV-G delivery (Yeaman et al., 2001; Grigoriev et al., 2007). siRNA depletion of *EXOC7* decreased the efficiency of *ts*VSV-G delivery to the plasma membrane (Liu et al., 2007). Studies into post-Golgi carriers and thus secretion have been hampered by the lack of model systems to study the kinetic process. The use of quantitative RUSH assays in this study allows this route to be studied with kinetics that better resemble endogenous trafficking. In addition, loss of the exocyst complex is lethal in cultured cells (Wang et al., 2015). To study KO of exocyst, we have used transient CRISPR-Cas9. This has two key advantages: It allows abrogation of the protein-of-interest to be studied in the appropriate phenotypic window and it avoids artefacts introduced by clonal selection after CRISPR-Cas9 gene editing. The combination of kinetic trafficking assays

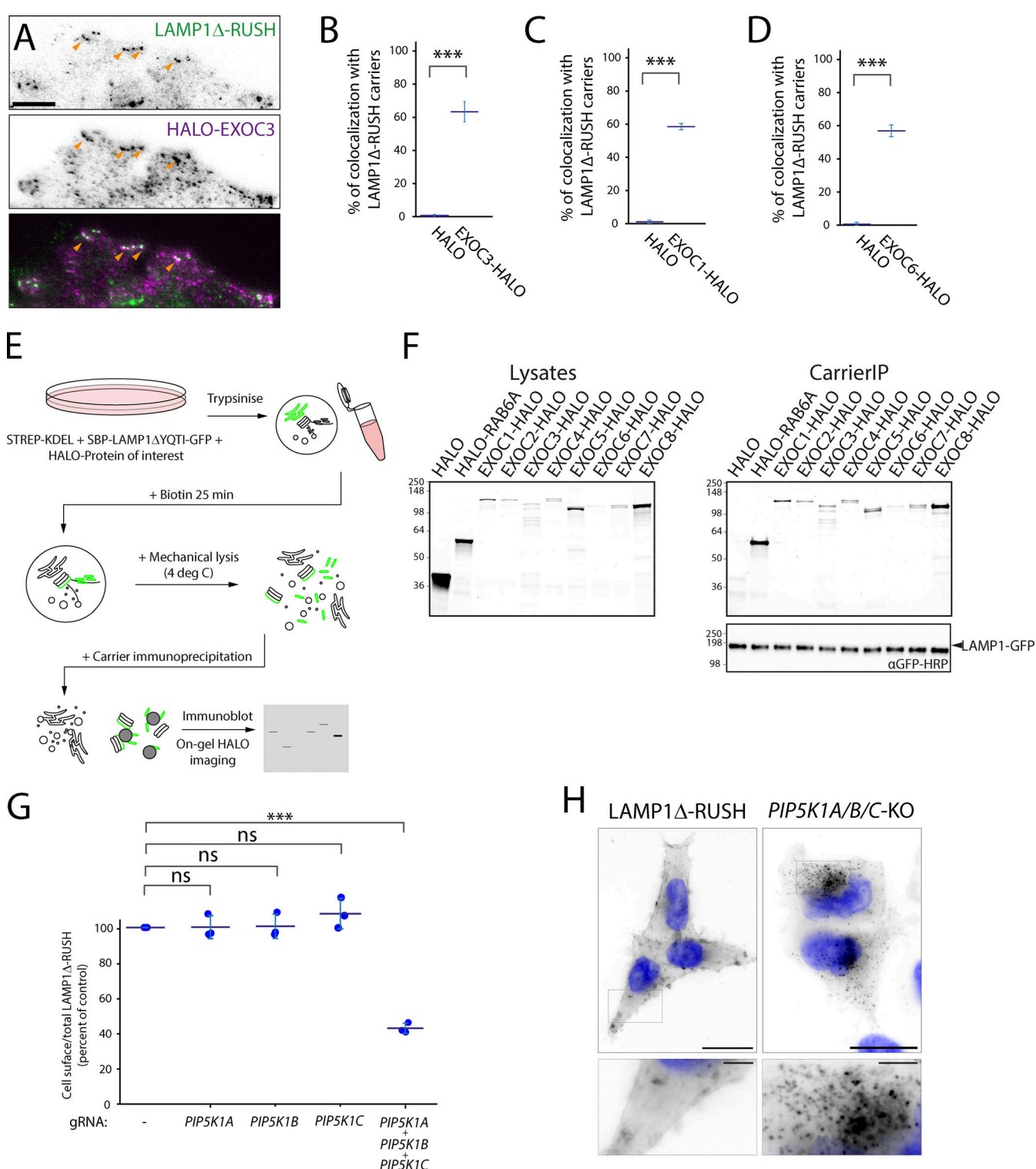

Figure 4. **Exocyst is recruited to post-Golgi carriers and multiple associated proteins are essential. (A)** TIRF imaging of heterologous expression of *EXOC3-HALO* in EXOC3-KO LAMP1Δ-RUSH cells. EXOC3-HALO (magenta) specifically co-localizes with LAMP1Δ-RUSH (green) carriers near the plasma membrane. Orange arrowheads indicate co-localizing structures. From Video 9. Scale bar: 5 μm. **(B)** Percent of LAMP1Δ-RUSH carriers positive for EXOC3-HALO, untagged HALO as a control, three biological repeats (total carriers quantified within 3 μm of plasma membrane edge = 298), statistical analysis = two-tailed *t* test. **(C)** Percent of LAMP1Δ-RUSH carriers positive for EXOC1-HALO, untagged HALO as a control, three biological repeats (total carriers quantified within 3 μm of plasma membrane edge = 228), statistical analysis = two-tailed *t* test. **(D)** Percent of LAMP1Δ-RUSH carriers positive for HALO-EXOC6, untagged HALO as a control, three biological repeats (total carriers quantified within 3 μm of plasma membrane edge = 206), statistical analysis = two-tailed *t* test. **(E)** Schematic representation of RUSH carrierIP assay. **(F)** Gels containing resolved proteins from carrierIP assay denoting enrichment of exocyst subunits (HALO-tagged EXOC1, EXOC2, EXOC3, EXOC4, EXOC5, EXOC6, EXOC7, and EXOC8) in LAMP1Δ-RUSH post-Golgi carriers (LAMP1 immunoblot). Molecular weight markers are indicated in kD. **(G)** Cell-surface ratio quantification (flow cytometry) showing reduced amounts of LAMP1Δ-RUSH at the plasma

membrane after transient KO of PIP5K homologs and 35 min of biotin exposure. **(H)** Widefield imaging of LAMP1Δ-RUSH reporter in triple PIP5K1A/B/C KO cells 1 h after biotin addition. When compared to WT, KO cells show accumulation of post-Golgi LAMP1Δ-RUSH carriers. Scale bar: 20 μm; insert: 4 μm. Nucleus stain = DAPI. Error bar = SD of at least three independent experimental repeats. Student's *t* test was performed on data in B, C, and D, and Tukey's multiple comparisons test (HSD, FWER = 0.05) was performed on data in G. ***P ≤ 0.001.

and transient CRISPR-Cas9 thus provides new insights into the fundamental process of secretion.

We chose LAMP1ΔYQTI as a probe as it has been experimentally demonstrated to traffic directly to the cell surface through the biosynthetic secretory pathway (Chen et al., 2017), and thus can act as an orthologous validated marker of this pathway, as demonstrated in Figs. 1 and 2. Additionally, LAMP1 can tolerate a tag on the N and C termini without affecting the trafficking or kinetics (Chen et al., 2017; Fig. S4). Some studies suggest that LAMP1 traffics through the endolysosomal system to the lysosome (Cook et al., 2004; the so-called "direct pathway"), and we cannot rule out a non-detectable subset of the RUSH cargo taking this pathway. Nevertheless, with the LAMP1ΔYQTI-RUSH, we observed a significant amount of trafficking directly to the cell surface (Fig. 1) in line with other studies (Chen et al., 2017).

Using the RUSH system coupled with kinetic trafficking assays, super-resolution imaging, and TIRF microscopy, we are able to map the machinery of the secretory pathway from the Golgi apparatus to the plasma membrane. RAB6A is essential for budding of the carriers from the Golgi (Fig. 2), where they traffic along microtubules to the cell tips. After budding, we see the tubules acquiring ARHGEF10 and RAB8 through imaging (Fig. 2), which is suggestive of a Rab cascade as previously described (Grigoriev et al., 2011; Shibata et al., 2016). A RAB6A to RAB8A transition has also been described in the tethering and fusion of carriers through ELKS (Grigoriev et al., 2011). Across different species, both RAB8 and RAB11 have been described as EXOC6 interactors (Zhang et al., 2004; Guo et al., 1999; Wu et al., 2005; Luo et al., 2014). It has also been suggested that yeast homologues of RAB8 and RAB11 together with the exocyst complex can promote vesicle transport along the cytoskeleton through EXOC6 interaction with myosin type V (Lipatova et al., 2008; Jin et al., 2011). EXOC5 has been shown to bind GTP-ARF6 (Prigent et al., 2003), which has been recently proposed to mediate the recruitment of a PIP5 kinase (Maib and Murray, 2021 Preprint). In fact, three PIP5 kinases appear to act redundantly to convert the post-Golgi phospholipids (Fig. 4) and allow the recruitment of the exocyst complex, which tethers the carrier to the plasma membrane for the final fusion event. In mammalian cells, the PIP5 kinases are likely recruited to post-Golgi membranes by several GTPases that act redundantly with ARF6. Indeed, this study has started to uncover a complex cargo delivery system, where redundancy most likely marks every step.

Major work has been undertaken on the exact molecular timings of recruitment of exocyst prior to cell-surface fusion which show an exquisite order of complex assembly on carriers, upstream of SNARE complex activity, and prior to cell-surface fusion (Rivera-Molina and Toomre, 2013; Ahmed et al., 2018). Our work is consistent with the observation that the exocyst complex is recruited to secretory carriers near the plasma membrane, prior to their fusion with the plasma membrane (Ahmed et al., 2018).

ELKS has previously been identified as the molecular tether for secretory carriers (Fourriere et al., 2019). Although we observed ELKS localizing to hotspots on the plasma membrane, we did not see a phenotype of *ELKS*-KO on cell-surface delivery of LAMP1Δ-RUSH (Fig. S2). This does not rule out the role of ELKS in this process and it is likely that for certain cargoes or cell types ELKS has an essential role. Accordingly, we observe an increase in the phenotype when combining *ELKS* and *exocyst* KO (Fig. S2 F), indicating a potential functional redundancy, and notably see both ELKS and exocyst on the same hotspots where cell-surface fusion occurs. Indeed, in neurons, studies demonstrate that ELKS acts as a redundant scaffold protein at the active zone site and that when this structure is disrupted, synaptic vesicle fusion is impaired (Wang et al., 2016). Additionally, one of the cargos that ELKS has been shown to have a role in the secretion of is NPY, a soluble secreted cargo that takes the regulated secretory pathway in specialized cell types (Fourriere et al., 2019). Besides a structural role, ELKS has been further shown to capture RAB6 positive cargoes in golgin-like manner contributing to the establishment of a ready to fuse pool of synaptic vesicles at the active zone (Nyitrai et al., 2020). To fully understand the specific balance of roles between exocyst and ELKS will require further studies of other cargoes in specific cell types.

A number of exocyst subunits have been implicated in rare human genetic disorders. To date and to our knowledge, there are five disease-associated subunits, *EXOC8* (Coulter et al., 2020; Online Mendelian Inheritance in Man [OMIM]: 615283), *EXOC7* (Coulter et al., 2020; OMIM: 608163), *EXOC6B* (Girisha et al., 2016; Simsek-Kiper et al., 2022; OMIM: 607880), and *EXOC4* (Nihalani et al., 2019; OMIM: 608185) and *EXOC2* (Van Bergen et al., 2020; OMIM: 615329). Mutations in *EXOC8* and *EXOC2* are associated with neurodevelopmental disorders, and mutations in *EXOC6B* with skeletal abnormalities and mutations in *EXOC4* have been associated with nephrotic syndrome. In cell models, exocyst has shown to be essential, with single-cell lethality associated with stable KOs (Wang et al., 2015). The severity and rarity of the disorders associated with loss of exocyst subunits are likely linked to both the variety of cellular functions associated with exocyst as well as the fundamental nature of these processes, including secretion.

Exocyst was initially associated with basolateral vesicle trafficking to cell–cell contacts in polarized cells (Grindstaff et al., 1998; Lipschutz et al., 2000; Langevin et al., 2005; Yeaman et al., 2004; Andersen and Yeaman, 2010; Xiong et al., 2012; Blankenship et al., 2007; Oztan et al., 2007; Bryant et al., 2010), and has since been implicated in a variety of processes including cytokinesis (Chen et al., 2006; Fielding et al., 2005;

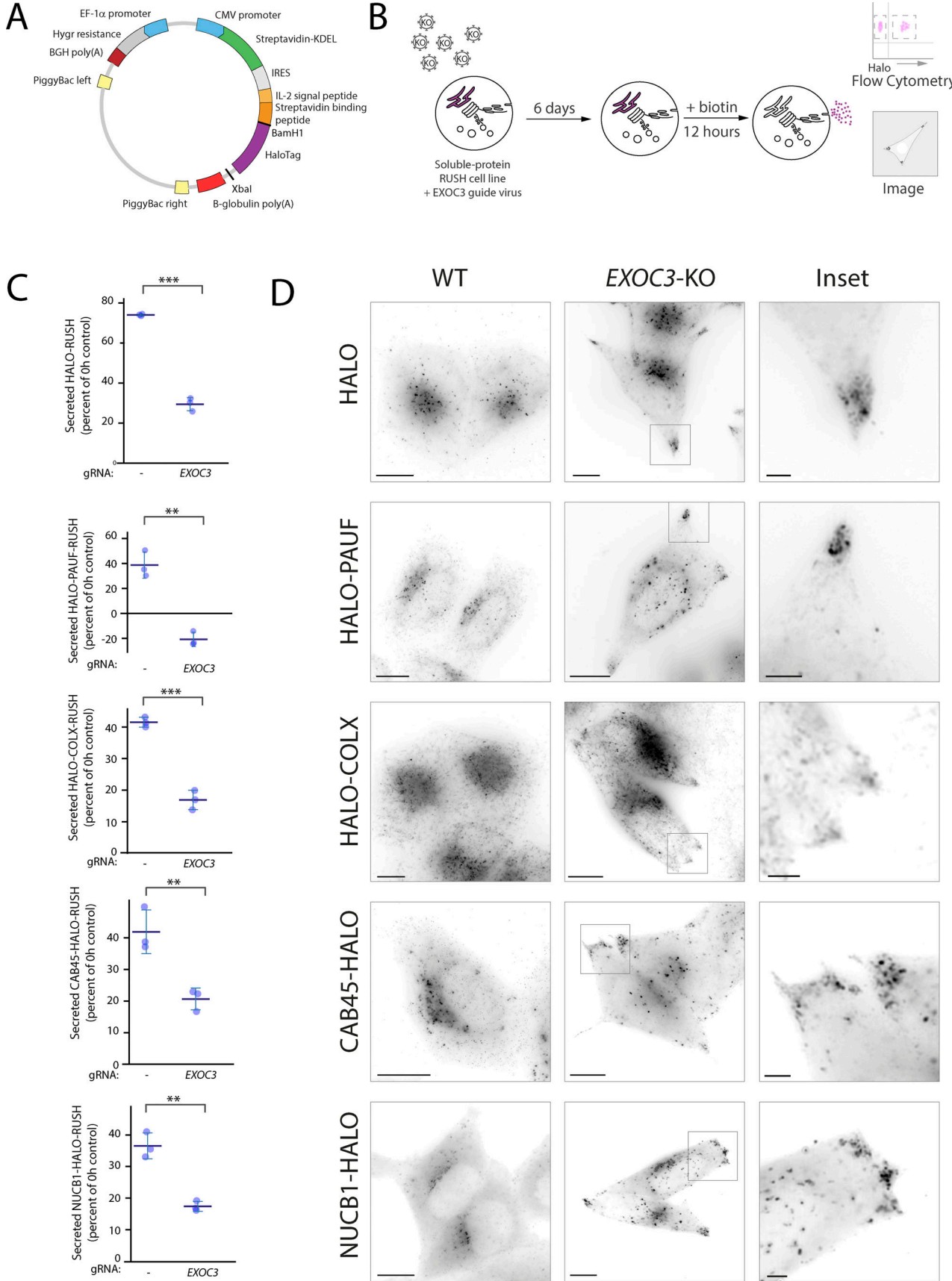

Figure 5. **Exocyst is essential for the secretion of various soluble cargoes. (A)** Schematic of novel PiggyBac transposon RUSH backbone. Genes encoding proteins of interest were either cloned upstream (BamHI) or downstream (XbaI) of HALO. **(B)** Schematic representation of guide infection followed by RUSH

and flow cytometry analysis/imaging of reporter protein of interest. **(C)** Flow cytometry–based analysis of protein secretion 12 h after biotin addition of indicated RUSH cargoes in WT and *EXOC3* KO cells. Each condition is normalized to 0 h control resulting in a percent of cargo secreted per sample and condition. Minimum of 30,000 cells per biological repeat, three repeats per sample (blue dots). Statistical comparison = two-tailed *t* test. **(D)** Live lattice-SIM imaging of RUSH cargo proteins in EXOC3 KO cells 12 h after biotin addition. When compared to WT, EXOC3 KO cells show substantial accumulation of carriers containing cargo proteins of interest near the plasma membrane. Scale bar: 10 µm; insert: 2 µm. Error bar = SD of at least three independent experimental repeats. Student's *t* test was performed on data in C. **P ≤ 0.01; ***P ≤ 0.001.

Neto et al., 2013), cell migration and tumor invasion (Rossé et al., 2006; Rosse et al., 2009; Spiczka and Yeaman, 2008; Assaker et al., 2010; Thapa et al., 2012; Lalli, 2009; Das et al., 2014), autophagy (Bodemann et al., 2011), lysosome secretion (Sáez et al., 2019), innate immune response following viral infection (Chien et al., 2006) and primary ciliogenesis (Rogers et al., 2004; Zuo et al., 2009; Feng et al., 2012), though in most cases, exocyst's role is directly linked to its exocytic function (Wu and Guo, 2015).

Here, we demonstrate that in addition to these roles, exocyst is essential for the fusion of constitutive secretory carriers. The regulation of exocyst and its recruitment to carriers is not fully understood.

There are a plethora of associated proteins that potentially allow for differential recruitment of exocyst to various carriers. These include the RALs (Maib and Murray, 2021 *Preprint*), ARF6 (Maib and Murray, 2021 *Preprint*), phospholipids (Maib and Murray, 2021 *Preprint*; Fig. 4, G and H), CDC42 (Zhang et al., 2001), RAB10 (Babbey et al., 2010), RAB11 (Takahashi et al., 2012), and RAB8A (Mei and Guo, 2018). In addition, *EXOC3* has three homologues, *EXOC3L1*, *EXOC3L2*, and *EXOC3L4*; and *EXOC6* has *EXOC6B*. For example, in this study, EXOC6 acts redundantly with EXOC6B for constitutive cargo delivery; however, the absence of EXOC6B alone is sufficient to cause a skeletal disorder in humans (Girisha et al., 2016; Simsek-Kiper et al., 2022). In addition, exocyst subunits have a differential tissue expression in various metazoa (Thisse et al., 2004; Mehta et al., 2005), and there are a number of functional sub-complexes ascribed to specific cellular functions, including the existence of sub-complex 1 (EXOC1–4) and 2 (EXOC5–8; Ahmed et al., 2018), as well as specialized sub-complexes including an EXOC8-depedent sub-complex (Bodemann et al., 2011), an EXOC2-EXOC8 containing sub-complex (Moskalenko et al., 2003; Jin et al., 2005), and a specialized role for EXOC7 (Zhao et al., 2013) and EXOC5 (Lipschutz et al., 2000). Which combination of these sub-complexes, homologues, or associated proteins provide specificity, redundancy, or regulation of exocyst is not fully understood, but could potentially explain the widespread function of the complex with discrete specificities.

## Materials and methods
### Antibodies and other reagents
The following primary antibodies were used for Western blot in this study: mouse anti-GFP HRP conjugate antibody (GG4-2C2.12.10; 1:5,000; 130-091-833; Miltenyi Biotec), rabbit anti-EXOC1 antibody (1:5,000; ab118798; Abcam), rabbit anti-EXOC3 antibody (EPR10812; 1:5,000; ab156568; Abcam), rabbit anti-ELKS antibody (EPR13777; 1:5,000; ab180507; Abcam), mouse

anti-LAMP1 antibody (H4A3; 1:10,000; ab25630; Abcam), rabbit anti-TIMP2 antibody (D18B7; 1:1,000; 5738; Cell Signaling Technology), rabbit anti-Cystatin C antibody (EPR4413; 1:5,000; ab109508; Abcam), rabbit anti-PSAP antibody (1:1,000; HPA004426; Atlas), mouse anti-GM130 antibody (1:1,000; 610822; BD Transduction Laboratories), and rabbit anti-GAPDH HRP conjugate antibody (D16H11; 1:1,000; 8884; Cell Signaling Technology). Goat HRP-conjugated secondary antibodies (1:5,000) were purchased from Abcam (anti-mouse: ab205719; anti-rabbit: ab205718), and goat-anti-human IgG secondary antibody IRDye 800CW (1:15,000; 926-32232) from LI-COR. The prokaryote expression vector encoding an anti-GFP mCherry nanobody (a gift from Martin Spiess, #109421; Addgene plasmid) and pOPINE GFP nanobody:HALO:His6, encoding anti-GFP HALO nanobody (a gift from Lennart Wirthmueller, #111090; Addgene plasmid) were expressed in bacteria and respectively GST- and His-purified in-house. The following cell-permeable dyes were obtained from these vendors: 646 HALO Dye (GA112A; Promega) and DAPI (D21490; Invitrogen).

The following antibiotics were used to select the newly generated stable cell lines: geneticin (875 mg/ml; 10131035; Life Technologies), hygromycin B (250 mg/ml; 10687010; Invitrogen), puromycin (1 µg/ml; A1113803; Gibco), and blasticidin (150 mg/ml; A1113903; Gibco).

The following chemicals were used in this work: dexamethasone (D4902; Sigma-Aldrich), biotin (B4501; Sigma-Aldrich), insulin (I5500; Sigma-Aldrich), 3-Isobutyl-1-methylxanthine (IBMX; I5879; Sigma-Aldrich), polybrene (TR-1003; Sigma-Aldrich).

### Plasmids
SBP-GFP-LAMP1ΔYQTI was PCR amplified from the original backbone SBP-GFP-LAMP1 (Chen et al., 2017; a generous gift from Juan Bonifacino, National Institute of Child Health and Human Development, National Institutes of Health) and Gibson assembled (E2621L; New England Biolabs) to the pEGFP-C1 (Clontech) vector backbone in between the AgeI/HindIII restriction sites. The SBP-LAMP1ΔYQTI-GFP was then obtained by a two-step assembly that first removed GFP and then re-cloned its PCR product downstream of LAMP1ΔYQTI.

Strep-KDEL was PCR amplified from Strep-KDEL_SBP-mCherry-GPI (#65295; Addgene plasmid) and assembled to a BamHI/PsrI digested TtTMPV-Neo viral backbone (#27993; Addgene plasmid). The neomycin resistance gene was then replaced with a hygromycin B encoding sequence.

pHALO-C1 and pHALO-N1 were generated by replacing the eGFP in Clontech vectors with HALO using Gibson assembly (E2621L; New England Biolabs). HALO-ARHGEF10, EXOC1-HALO, EXOC2-HALO, EXOC3-HALO, EXOC4-HALO, EXOC5-HALO, EXOC6-HALO, HALO-EXOC6, EXOC7-HALO, EXOC8-HALO, HALO-

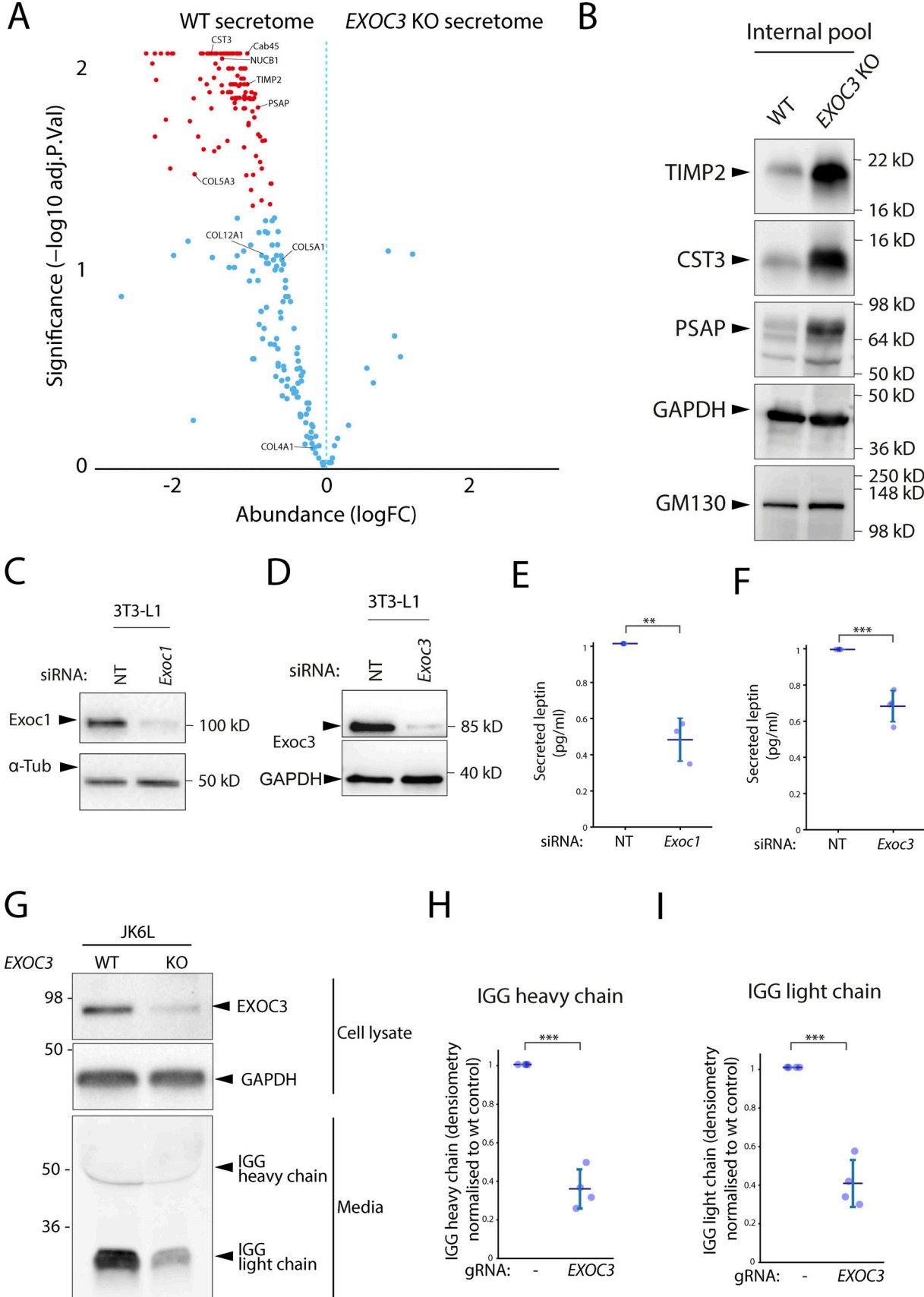

Figure 6. **Exocyst is essential for secretion of a broad array of endogenous soluble secreted proteins. (A)** Volcano plot from the mass spectrometry data on soluble proteins secreted over a period of 6 h by WT HeLa cells demonstrates that EXOC3 KO causes a dramatic and significant decrease of protein secretion

in these cells. Significantly different proteins are shown in red (P ≤ 0.01). **(B)** Immunoblot showing intracellular accumulation of selected proteins with downregulated secretion in EXOC3 KO cells. **(C)** Immunoblot confirming downregulation of *Exoc1* in a siRNA-treated population of mouse 3T3-L1 adipocytes. **(D)** Immunoblot confirming downregulation of *Exoc3* in a siRNA-treated population of mouse 3T3-L1 adipocytes. **(E)** ELISA quantification of secreted leptin from 3T3-L1 adipocytes siRNA treated with non-targeting (NT) or *Exoc1*. Results normalized to non-targeting control. **(F)** ELISA quantification of secreted leptin from 3T3-L1 adipocytes treated with non-targeting (NT) or *Exoc3* siRNA. Results normalized to non-targeting control. **(G)** Immunoblot showing that down-regulation of EXOC3 in a population of human JK6L lymphocytes correlates with a significant decrease in IgG secretion over a period of 24 h. **(H)** Quantification of secreted heavy chain IgG over four independent experiments described in F. **(I)** Quantification of secreted light chain IgG over four independent experiments described in F. Error bar = SD of at least three independent experimental repeats. Two-tailed *t* test was performed on data in E, F, H, and I. **P ≤ 0.01; ***P ≤ 0.001.

ELKS, and HALO-EXOC6 were cloned by Gibson assembly using a synthetic, codon optimized version of each gene of interest (Integrated DNA Technologies), cloned in-frame upstream or downstream of HALOTag in pHALO-C1 or pHALO-N1. HALO-RAB6A and HALO-RAB8A were cloned in similar manner, except the gene sequences were PCR amplified from pEGFP-RAB6A (a kind gift from Juan Bonifacino, National Institute of Child Health and Human Development, National Institutes of Health) and pEGFP-RAB8A (Matsui et al., 2011; a gift from Mitsunori Fukuda, Tohoku University, Sendai, Japan), respectively. Point mutations were introduced through Q5 Site-Directed Mutagenesis Kit (M0554S; New England Biolabs).

Microtubules were visualized by expressing a plasmid containing β-tubulin-mCherry (#175829; Addgene plasmid). To generate a stable KO cell line, guide RNAs targeting ELKS (ERC1) were cloned into pSpCas9 (BB)V2.0 (#62988; Addgene plasmid; Ran et al., 2013), using the BbsI restriction sites.

For transient KO cells, guide RNAs targeting a gene of interest (Table S1) were cloned into pKLV-U6gRNA (BbsI)-PGKpuro2ABFP (#50946; Addgene plasmid), using the BbsI restriction sites, as described above. The IDT Alt-R CRISPR-Cas9 guide RNA tool was used to custom design two guide sequences per gene of interest. Cas9 viral expression backbone was a kind gift from Paul Lehner as well as the packaging vectors pMD.G and pCMVR8.91.

To generate the piggybac-RUSH constructs, we first created a piggybac-CMV-StrepKDEL-IRES-SBP-HALO. To achieve this, we Gibson assembled a PCR product containing CMV-StrepKDEL-IRES-SBP (#65295; amplified from Addgene plasmid) to the SalI/MluI digested piggybac backbone (a generous gift from Jonathon Nixon-Abell, Cambridge Institute for Medical Research [CIMR], University of Cambridge, Cambridge, UK) and then digested the result with BamHI to assemble in a HALOTag flanked by BamHI and XbaI restriction sites. Gibson assembly was used to further clone other cDNAs of interest into this piggybac-RUSH-HALO vector. A PCR product of ColX (#110726; from Addgene) and a synthetic gene containing PAUF (Integrated DNA Technologies) were cloned into the XbaI site downstream of HALO. PCR products containing CAB45 and NUCB1 (generous gifts from Liz Miller), and TNFa (#166901; from Addgene) were cloned into the BsrGI/BamHI sites upstream of HALO. Plasmids and primers used in this work are available upon request. All constructs were sequenced to verify their integrity.

**Cell lines**

HeLa cells were already available in the lab; 3T3-L1 fibroblasts, originally from Howard Green (Harvard Medical School, Boston, MA), were a gift from David James (University of Sydney, Australia); JK6L myeloma cells were a kind gift from Jonathan Keats (Translational Genomics Research Institute, USA). and Lenti-X 293T cells were obtained from Takara Bio (632180). JK6L cells were maintained in RPMI 1640 (21875034; Gibco) supplemented with 10% FBS (F7424; Sigma-Aldrich), 1% GlutaMAX (35050061; Gibco), and MycoZap Plus-CL (VZA-2012; Lonza). They were kept in a humidified 5% $CO_2$ atmosphere at 37°C. All other cell lines were grown in DMEM high glucose (D6429; Sigma-Aldrich) supplemented with 10% FBS (F7424; Sigma-Aldrich) and MycoZap Plus-CL (VZA-2012; Lonza), and were kept at 37°C in a humidified 5% $CO_2$ atmosphere (10% $CO_2$ for 3T3-L1 fibroblasts).

For adipocyte differentiation, confluent 3T3-L1 fibroblasts were treated with 220 μM dexamethasone (D4902; Sigma-Aldrich), 100 ng/ml biotin (B4501; Sigma-Aldrich), 2 mg/ml insulin (I5500; Sigma-Aldrich), and 500 mM IBMX (I5879; Sigma-Aldrich) in complete DMEM. After 3 d, media was replaced with fresh complete DMEM supplemented with 2 mg/ml insulin only, and after another 3 d (day 6 of differentiation), the media was then replaced with complete DMEM alone and subsequently changed every 48 h.

A HeLa cell line, stably expressing the Strep-KDEL, was generated by infection with retroviral particles containing the Strep-KDEL plasmid followed by hygromycin B selection and subsequent single-cell clonal isolation.

Both HeLa stable cell lines expressing either SBP-LAMP1-ΔYQTI-GFP or SBP-GFP-LAMP1ΔYQTI were created by transient transfection of corresponding vector backbones using Lipofectamine 2000 (11668019; Invitrogen), followed by selection with geneticin and single clonal isolation using live cell sorting. For SBP-LAMP1ΔYQTI-GFP two cell clones were selected: one low expression suitable for biochemical studies (i.e., carrierIP) and another with higher expression to image. As for the SBP-GFP-LAMP1ΔYQTI cell line, a medium-to-low expression clone was selected, suitable for the flow cytometry LAMP1 cell-surface assay.

Lenti-X 293T cells were used to package the pKLV-puro vectors encoding plasmid/guide RNAs into lentiviral particles as previously described (Pirona et al., 2020). Viral supernatants were harvested after 48 h, filtered through a 0.45 μm filter, and when needed, concentrated down 10 times using the Lenti-X Concentrator (631232; Takara Bio). Supernatants were kept at –80°C prior to being directly applied to target cells, which were then spun at 700 × *g* for 1 h at 37°C. When necessary, cells were transiently selected with the appropriate antibiotic 48-h after transduction.

Stable Cas9 expressing cell lines were generated by infecting target cells with lentiviral particles carrying Cas9 plasmid DNA (generous gift from Paul Lehner) followed by selection for blasticidin antibiotic resistance. Cas9 expression on >96% of the cell population was further confirmed through flow cytometry analysis, by testing loss of cell-surface expression of β-2 microglobulin, upon transduction with lentiviral particles containing a β-2 microglobulin targeting single guide RNA, using a mouse monoclonal anti-B2M antibody (a generous gift from Paul Lehner).

## RUSH and flow cytometry

$1 \times 10^6$ LAMP1Δ-RUSH HeLa cells were resuspended in DMEM supplemented with 25 mM Hepes (25 mM; 15630080; Gibco) and transferred to 1.5 ml microcentrifuge tubes. Tubes were kept at 37°C in a heating block (DB200/2; Techne), and D-biotin (B4501; Sigma-Aldrich) at a final concentration of 500 μM was added at each time point (for the majority of experiments 35 min). Cells were then incubated on ice for 5 min to stop RUSH and subsequently spun down (4°C, 500 $g$, 5 min) to remove the supernatant. All downstream manipulations were either on ice or at 4°C. Following this, cells were incubated with an mCherry anti-GFP nanobody (10 μg/ml in PBS) for 1 h, and then washed two times with 500 μl of PBS. Finally, cells were filtered using the Cell-Strainer capped tubes (352235; FALCON). A minimum of 30,000 cells per sample was analyzed using an LSRFortessa cell analyzer (BD Biosciences), gating for GFP-positive cells (indicative of LAMP1Δ-RUSH expression), plus any other concomitant fluorophore when appropriate. Data were analyzed using FlowJo software (v10.8.1). RUSH was inferred by single-cell plotting the relative intensity of mCherry (LAMP1 at the PM) over that of GFP (LAMP1Δ available to RUSH).

## CRISPR KO

Stable ELKS1 KO HeLa cells were generated using the CRISPR-Cas9 system (Ran et al., 2013). Plasmids containing Cas9 and single-guide RNAs were transfected (Lipofectamine, 2000; 11668019; Invitrogen) into the LAMP1Δ-RUSH cell line according to manufacturers' instructions. After 48 h, cells were selected with puromycin for a week before being single-cell sorted to establish clonal cell lines. Candidate clones were validated by immunoblotting to confirm the loss of ELKS.

Transient CRISPR KO was achieved by transducing previously established stable Cas9 cell lines. These include HeLa-Cas9, LAMP1ΔYQTI-GFP-Cas9, SBP-GFP-LAMP1ΔYQTI-Cas9, and JK6L-Cas9. Briefly, $25 \times 10^3$ LAMP1Δ-RUSH cells were transduced with 200 μl of lentiviral supernatant or 30 μl of 10× concentrated lentiviral particles in a 48-well plate. 48-h post-transduction cells were replated in a 6-well plate and incubated for an extra 4 d. On day 6, cells were detached with trypsin, and each well split into two, for two RUSH time points (0 and 35 min), followed by flow cytometry analysis. Depletion of target protein was verified by immunoblotting and/or qPCR analysis. For some experiments, infections were scaled up accordingly. To recover the cell secretion phenotype, day 5 transient KO cells were electroporated (Gene Pulser Xcell Total System; Bio-Rad) with a guide-resistant plasmid encoding the corresponding protein of interest. In brief, $1 \times 10^6$ cells were electroporated with 1 μg of plasmid DNA in a

final volume of 200 μl of Opti-MEM I (31985062; Gibco), in a 2 mm electroporation cuvette (Z706086; Sigma-Aldrich), using the manufacturer's settings for HeLa cells, and plated in complete DMEM containing 646 HALO Dye (20 nM; GA112A; Promega) immediately after. The following day, cells were lifted with trypsin, RUSHed for 35 min, and analyzed by flow cytometry as described above.

For *EXOC3* KO, $400 \times 10^5$ JK6L-Cas9 cells were transduced with 30 μl of 10× concentrated lentiviral particles in the presence of 0.8 μg/ml polybrene (TR-1003; Sigma-Aldrich) and a final volume of 200 μl of complete DMEM. This was carried out in a 48-well plate, without any centrifugation. The next day, cells were replated in fresh complete RPMI and moved to a 6-well plate where they were incubated until the last day of the experiment. On day 2, puromycin was added (5 μg/ml) to select transduced myeloma cells. On day 5, cells were counted and seeded in a 96-well plate at a density of $2 \times 10^6$/ml in complete fresh RPMI. 24 h later both the media and cell fractions were collected for immunoblot analysis of secreted IgG.

## siRNA

siRNA was delivered to differentiated adipocytes (6–7 d after differentiation) using the TransIT-X2 dynamic delivery system (MIR 6004; Mirus Bio). Opti-MEM I (31985062; Gibco) and TransIT-X2 were mixed together (ratio 25/1). siRNA (siGENOME SMARTpool #69940; Dharmacon Horizon) was added to the Opti-MEM/TransIT-X2 mix such that the final concentration per well was 100 nM, mixed gently by pipetting, and incubated at room temperature for 30 min. 450 μl of cell suspension was added to the Opti-MEM/TransIT-X2/siRNA mix and mixed well. 570 μl of the resulting mixture was seeded into one well of a 24-well plate. Cells were fed with fresh culture media 24 h after transfection and incubated at 37°C, 10% $CO_2$ for a further 72 h. 96 h after initial transfection, cells were transfected with siRNA for a double-hit knockdown. Transfection mix was prepared as described above, except media alone was added to the Opti-MEM/TransIT-X2 mix instead of cell suspension. 570 ml of transfection mix was added to each well of adhered cells. Culture media was replaced with fresh media 24 and 72 h after transfection. 96 h after the second transfection, the culture media was harvested for analysis and cells were lysed for immunoblotting. Leptin present in the culture media was detected using the MesoScale Discovery mouse leptin kit (MesoScale Discovery, K152BYC-2), and the resulting amounts were normalized to the non-targeting control.

## carrierIP

### Preparation of GFP nanobody magnetic dynabeads

Magne HALOTag Beads (G7281; Promega) were incubated with purified HIS-HALO-GFP nanobody protein in 20 mM Hepes, pH 7.5, 150 mM NaCl buffer overnight at 4°C. The beads were washed in the same buffer and stored at 4°C in 50 mM Hepes, pH 7.5, 0.15 M NaCl, 15% (vol/vol) glycerol, 0.05% NaN3 (50% slurry).

### carrierIP

$1.75 \times 10^6$ SBP-LAMP1ΔYQTI-GFP cells were seeded in a 10-cm plate (per condition). 48 h later, the cells were transfected with HALO-C1, HALO-RAB6a, EXOC1-HALO, EXOC2-HALO, EXOC3-

HALO, EXOC4-HALO, EXOC5-HALO, EXOC6-HALO, EXOC7-HALO, and EXOC8-HALO plasmids (Lipofectamine 2000; 11668019; Invitrogen) according to manufacturers' instructions. 4 h after transfection, media was replaced with fresh complete DMEM containing 646 HALO Dye (20 nM; GA112A; Promega). The next day, cells were washed twice with 10 ml PBS and detached with 3 ml of trypsin. Cells were collected with 2 times 10 ml of DMEM, spun down (4°C, 500 × $g$, 10 min), and resuspended in 20 ml of DMEM supplemented with 25 mM Hepes (25 mM; 15630080; Gibco). Tubes were kept at 37°C in a water bath and D-biotin (B4501; Sigma-Aldrich) was added at a final concentration of 500 μM. Cells were incubated for 35 min at 37°C and then placed on ice; 20 ml of ice-cold PBS was added immediately to stop RUSH. Cells were spun down (4°C, 500 × $g$, 10 min) to remove the supernatant, washed with 10 ml PBS, and spun down again. Each pellet was resuspended with 1 ml of cytosol buffer (25 mM Hepes, 125 mM potassium acetate, 5.4 mM glucose, 25 mM magnesium acetate, adjust to pH 7, and add fresh 100 μM EDTA and 1× protease inhibitors), and the cells were homogenized with 25 strokes on ice using a cell homogenizer (8 micron bead; Isobiotec). Cell homogenates were spun at 3,000 × $g$ for 10 min at 4°C to remove cell debris. Supernatants were collected and 15 μl was saved in 15 μl of Laemmli buffer for the final gel. The rest of the supernatants were incubated for 15 min at 4°C with 10 μl of GFP-Trap Agarose beads (gta-10; Chromotek) to remove free GFP and spun at 1,200 × $g$ for 1 min at 4°C. Supernatants were then incubated with 30 μl of homemade GFP nanobody magnetic dynabeads for 15 min at 4°C to trap LAMP1Δ-RUSH carriers. The magnetic beads were washed two times with 1 ml of cytosol buffer (one quick wash and another for 5 min at 4°C) and 50 μl of 2× Laemmli buffer. Samples were boiled for 10 min at 95°C, and 10 μl of the lysates and 20 μl of the IP samples were resolved on a gradient Tris-Glycine acrylamide gel. The fluorescence of the 646-HaloTag ligand was directly imaged in the ChemiDoc Imaging System (Bio-Rad). LAMP1Δ-GFP was detected by immunoblotting GFP.

## Microscopy

For immunofluorescence microscopy, 40 × 10³ day 6 transduced cells were plated onto Matrigel-coated (1:100 in complete DMEM; 354277; Corning) glass coverslips (400-03-19; Academy) in the presence of puromycin. The next day, coverslips were fixed with cytoskeletal fixing buffer (300 mM NaCl, 10 mM EDTA, 10 mM Glucose, 10 mM MgCl₂, 20 mM PIPES, pH 6.8, 2% Sucrose, 4% PFA) for 15 min and DAPI stained (300 nM; D21490 Invitrogen) for 5 min. PBS was used to wash cells in between all steps, and coverslips were mounted in ProLong Gold (P36930; Life Technologies).Standard epifluorescent images were obtained with an Axio Imager.Z2 microscope (Zeiss) equipped with an Orca Flash 4.0 camera (Hamamatsu), an HXP 120V light source, and a 100× NA 1.4 Plan-Apochromat objective, all under the control of ZEN software (Zeiss).

For live-cell imaging, 9 × 10⁴ SBP-LAMP1ΔYQTI-GFP cells were plated onto Matrigel-coated glass coverslips (CB00250RAC; Menzel-Gläser). When necessary, FuGENE 6 was used to transfect constructs encoding proteins of interest the following day. 2 h after transfection, media was replaced with fresh complete DMEM containing 646 HALO Dye (20 nM; GA112A; Promega). The next

day, cells were imaged in an Elyra 7 with Lattice SIM² microscope (Zeiss) equipped with an environmental chamber (temperature controlled at 37°C, humidified 5% CO2 atmosphere), two PCO.edge sCMOS version 4.2 (CL HS) cameras (PCO), solid state diode continuous wave lasers, and a Zeiss alpha Plan-Apochromat 63×/1.46 Oil Corr M27 objective for TIRF imaging and a Zeiss Plan-Apochromat 63×/1.4 Oil DIC M27 used for lattice-SIM, all under the control of ZEN black software (Zeiss). SBP-LAMP1ΔYQTI-GFP carriers (GFP) were assessed for co-localization with HALO-tagged protein of interest, with expression of cytosolic HALO as a control. The percent of LAMP1Δ-RUSH carriers positive for HALO-tagged protein of interest, among three biological repeats carried out between 35–45 min of biotin exposure, was then plotted and followed by statistical analysis ($t$ test).

## Secretomics

160 × 10³ HeLa-Cas9 cells were transduced with 1 ml of EXOC3 lentiviral supernatant (total volume 2 ml) in a 6-well plate. The following day, media was replaced with fresh complete DMEM. 2 d later cells were replated into a 15-cm dish along with complete DMEM supplemented with puromycin, for EXOC3 KO condition. At infection day 7, both control and selected EXOC3 KO cells were washed three times with 30 ml PBS Ca⁺Mg⁺ (one quick wash, another for 5 min at 37°C, and a third quick one; D866; Sigma-Aldrich) and incubated with 15 ml of FBS free DMEM for 6 h. The media was then collected and cooled on ice, the cells lifted, counted, and sample buffer was added for a final concentration of 5 × 10⁶ cells/ml in 1× Laemmli buffer. Media was spun at 500 × $g$ for 5 min at 4°C to exclude floating cells. Supernatant was then centrifuged at 4,000 × $g$ for 15 min at 4°C to exclude other cellular debris. The remaining media was then concentrated down to 500 μl using a falcon tube sized 3 kD amicon column (UFC900308; Merck) spun at 4,000 × $g$ and 4°C for ~1 h. Concentrated media was analyzed on a Q Exactive Orbitrap Mass Spectrometer (Thermo Fisher Scientific), and protein hits were filtered using a custom R script to select for proteins with a signal peptide, without a transmembrane domain and without a GPI anchor as per Uniprot annotations.

## Immunoblotting

Total denatured cell extracts (100,000 cells) were resolved on a gradient Tris-Glycine acrylamide gel and transferred to a polyvinylidene difluoride membrane, which was blocked with 5% skimmed milk and incubated with a primary antibody overnight at 4°C, followed by a 1-h incubation with a secondary antibody at room temperature. Membranes were washed with phosphate-buffered saline with 0.1% Tween 20 in between steps, and finally, their immunoreactivity was visualized using Clarity (1705061; Bio-Rad) or WesternBright Sirius (K-12043-D10; Advansta) ECL substrate, in the ChemiDoc Imaging System (Bio-Rad). Bands were quantified using Image Lab software, version 6.1 (Bio-Rad), and GAPDH was used as a loading control.

## RNA extraction and qPCR

Total RNA was isolated from cultured WT or KO cells (35-mm dish), using the RNeasy Mini Kit (74104; Qiagen) according to the manufacturer's instructions. RNA quantification was performed

using NanoDrop One (Thermo Fisher Scientific). Strand cDNA was generated by priming 1 µg of total RNA with an oligo (dT) 16/random hexamers mix, using the High-Capacity RNA-to-cDNA Kit (4387406; Applied Biosistems), following manufacturer's instructions. cDNA templates were diluted 10-fold and 1 µl was used with specific oligos spanning 2 exons (Table S3) along with the PowerUp SYBR Green Master Mix (A25741; Applied Biosistems) for the qPCR reaction. All reactions were performed in three technical replicates using the CFX96 Touch Real-Time PCR Detection System (Bio-Rad). Data was analyzed according to the $2^{-\Delta\Delta CT}$ method (Livak and Schmittgen, 2001).

### Statistical analysis

Statistical analysis was performed using GraphPad Prism v.5 or Python 3.7 and statistical significance was considered when $P < 0.05$. Comparisons were made using Student's $t$ test or Multiple Comparison of Means—Tukey Honestly Significant Difference (HSD), family-wise error rate (FWER) = 0.05. Unless stated, all quantitative data are expressed as mean ± SD of at least three independent experiments.

### Online supplemental material

Fig. S1 shows a GFP immunoblot demonstrating increased LAMP1Δ-RUSH glycosylation over time after biotin addition. Fig. S2 shows ELKS and its associated proteins and/or their homologs are not necessary for LAMP1Δ-RUSH post-Golgi tubule fusion. Fig. S3 shows additional subunits EXOC1 and EXOC6 are directly recruited to post-Golgi carriers and PIP5K activity is essential for carrier delivery. Fig. S4 shows either C- or N-terminally tagged LAMP1Δ-RUSH trafficking with comparable kinetics to the cell surface. Fig. S5 shows the different soluble RUSH cargos being effectively retained in the ER. Table S1 lists gRNA sequences used for this work. Table S2 details the post-analysis secretomics data set. Table S3 lists all oligos used for qPCR. Video 1 shows the type-1 membrane spanning RUSH reporter LAMP1Δ-GFP trafficking from the ER (0′), to the Golgi apparatus (15′), and then directly to the plasma membrane (30′–65′). Video 2 shows LAMP1Δ-RUSH leaving the Golgi apparatus in tubular carriers ~35 min after biotin addition. Video 3 shows LAMP1Δ-RUSH carriers (green) moving along the microtubular network (magenta). Video 4 shows the plasma membrane TIRF plane where LAMP1Δ-RUSH post-Golgi tubules can be seen fusing. Video 5 shows heterologous expression of *HALO-RAB6A* in WT HeLa cells with RAB6A present at the Golgi and in tubular structures that bud off and travel toward the plasma membrane. Video 6 shows HALO-RAB6A (magenta) co-localizing with LAMP1Δ-RUSH tubules (green) leaving the Golgi and moving toward the plasma membrane (~35 min after biotin addition). Video 7 shows co-localization of HALO-ARHGEF10 (magenta) with LAMP1Δ-RUSH carriers (green) traveling toward the plasma membrane (~35 min after biotin addition). Video 8 shows plasma membrane TIRF microscopy showing HALO-RAB8A (magenta) co-localizing with LAMP1Δ-RUSH carriers (green) near their fusion site. Video 9 shows co-localization of EXOC3-HALO (magenta) with LAMP1Δ-RUSH (green) carriers in EXOC3-KO LAMP1Δ-RUSH cells. Video 10 shows LAMP1Δ-RUSH carriers (green) fusing in plasma membrane

sites enriched for HALO-ELKS (magenta). Video 11 shows LAMP1Δ-RUSH carriers (green) co-localizing with EXOC1-HALO (magenta) after ~35 min in biotin. Video 12 shows LAMP1Δ-RUSH carriers (green) co-localizing with HALO-EXOC6 (magenta) after ~35 min in biotin.

## Acknowledgments

We thank Paul Lehner (Cambridge Institute of Therapeutic Immunology & Infectious Diseases, University of Cambridge, UK), Dick van den Boomen (Cambridge Institute of Therapeutic Immunology & Infectious Diseases, University of Cambridge, UK), and Liz Miller (MRC Laboratory of Molecular Biology, Cambridge, UK) for molecular biology reagents and advice. We thank Paul Luzio, Margaret Robinson, and Laura Pellegrini for reading the manuscript, helpful discussions, and advice.

This research was supported by the CIMR Flow Cytometry Core Facility and the CIMR Microscopy Committee. In particular, we wish to thank Matthew Gratian, Reiner Schulte, and Gabriela Grondys-Kotarba for their advice and support in imaging, flow cytometry, and cell sorting. D.C. Gershlick and D. Stalder are funded by a Sir Henry Dale Fellowship awarded to D.C. Gershlick from the Wellcome Trust/Royal Society (Grant 210481). C. Pereira is funded by a Biotechnology and Biological Sciences Research Council responsive mode grant (BB/W005905/1), a Wellcome Trust Institutional Strategic Support Fund awarded to D.C. Gershlick and an Isaac Newton Trust Research Grant D.J. Fazakerley and A.S. Shun-Shion were supported by a Medical Research Council (MRC) Career Development Award to D.J. Fazakerley (MR/S007091/1). M.A. Chapman and G. Anderson were supported by the MRC Toxicology Unit (MC_UU_00025/10). This work was funded by a Wellcome and UK Research and Innovation grant (grant 210481, MR/S007091/1). For the purpose of open access, the author has applied a Creative Commons Attribution (CC BY) license to any Author Accepted Manuscript version arising. Leptin secretion studies were supported by the MRC Metabolic Diseases Unit Mouse Biochemistry Laboratory [MC_UU_00014/5], Cambridge.

Author contributions: D.C. Gershlick and C. Pereira conceived of the project. The bulk of experiments were designed by C. Pereira, D. Stalder, and D. Gershlick and performed by C. Pereira and D. Stalder. G. Anderson, A.S. Shun-Shion, and D. Gershlick contributed with additional experimental work. J. Houghton and R. Antrobus performed and analyzed the proteomics data. A.S. Shun-Shion and D.J. Fazakerley performed and analyzed adipocyte data. Methodology was designed by M.A. Chapman, D. Gershlick, D.J. Fazakerley, C. Pereira, D. Stalder, and R. Antrobus. Additional resources were provided by M.A. Chapman and D.J. Fazakerley. The manuscript was written by D. Gershlick, C. Pereira, and D. Stalder and reviewed and edited by D.J. Fazakerley and M.A. Chapman along with other co-authors.

Disclosures: The authors declare no competing interests exist.

Submitted: 31 May 2022

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

## Supplemental material

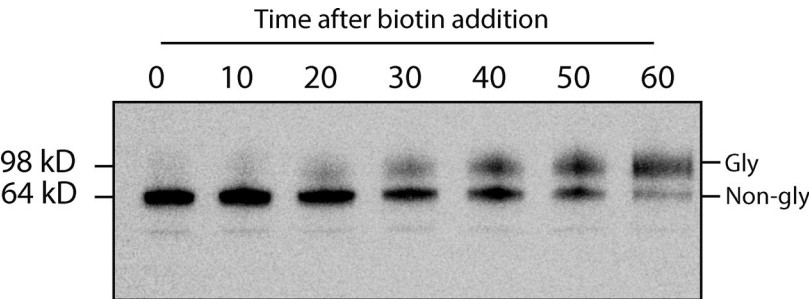

Figure S1. **LAMP1ΔYQTI-RUSH is glycosylated during RUSH assay.** GFP immunoblot showing increasing LAMP1Δ-RUSH glycosylation over time after biotin addition. Note that LAMP1Δ-RUSH starts showing glycosylation from 20 min onwards. Gly = glycosylated LAMP1Δ-RUSH, Non-gly = non-glycosylated LAMP1Δ-RUSH.

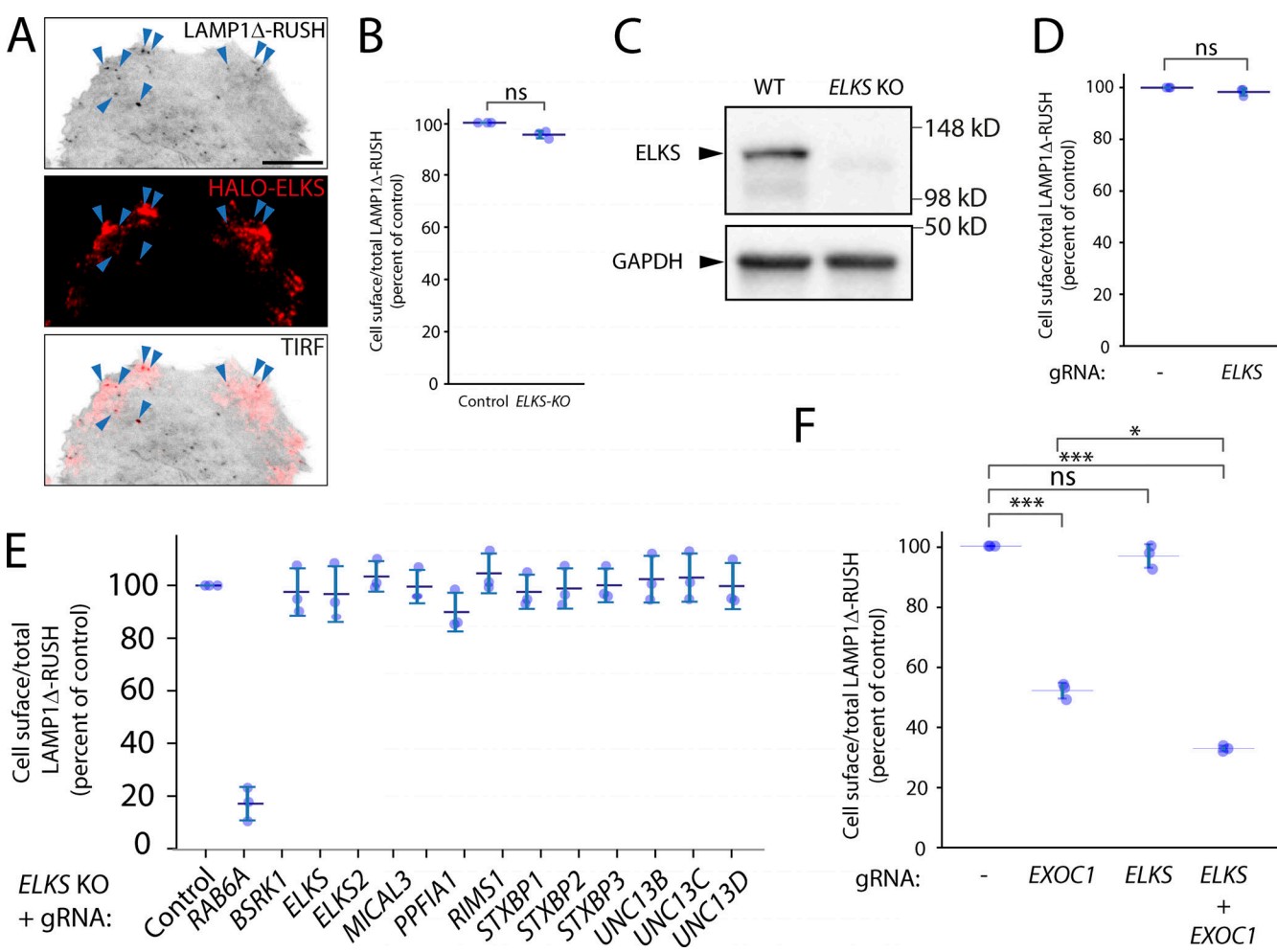

Figure S2. **ELKS and its associated proteins and/or homologs are not necessary for LAMP1Δ-RUSH post-Golgi tubule fusion. (A)** TIRF imaging of LAMP1Δ-RUSH (gray) cells expressing heterologous HALO-ELKS (red) mostly localized to fusion sites (blue arrowheads). From Video 10. Scale bar: 10 μm. **(B)** Cell-surface ratio quantification (flow cytometry) of LAMP1Δ-RUSH at the plasma membrane after stable ELKS KO and 35 min of biotin exposure. **(C)** Immunoblot confirming loss of ELKS in a stable LAMP1Δ-RUSH ELKS KO clonal cell line. **(D)** Cell-surface ratio quantification (flow cytometry) of LAMP1Δ-RUSH at the plasma membrane after transient ELKS KO and 35 min of biotin exposure. **(E)** Cell-surface ratio quantification (flow cytometry) of LAMP1Δ-RUSH at the plasma membrane after transient KO of known ELKS interaction partners and 35 min of biotin exposure. This experiment was carried out in a stable clonal ELKS KO cell line. **(F)** Cell-surface ratio quantification (flow cytometry) of LAMP1Δ-RUSH at the plasma membrane after transient KO of EXOC1, ELKS, and combined. Error bar = SD of at least three independent experimental repeats. Two-tailed *t* test was performed on data in B and D, and Tukey's multiple comparisons test (HSD, FWER = 0.05) was performed on data in F. *P ≤ 0.05; ***P ≤ 0.001.

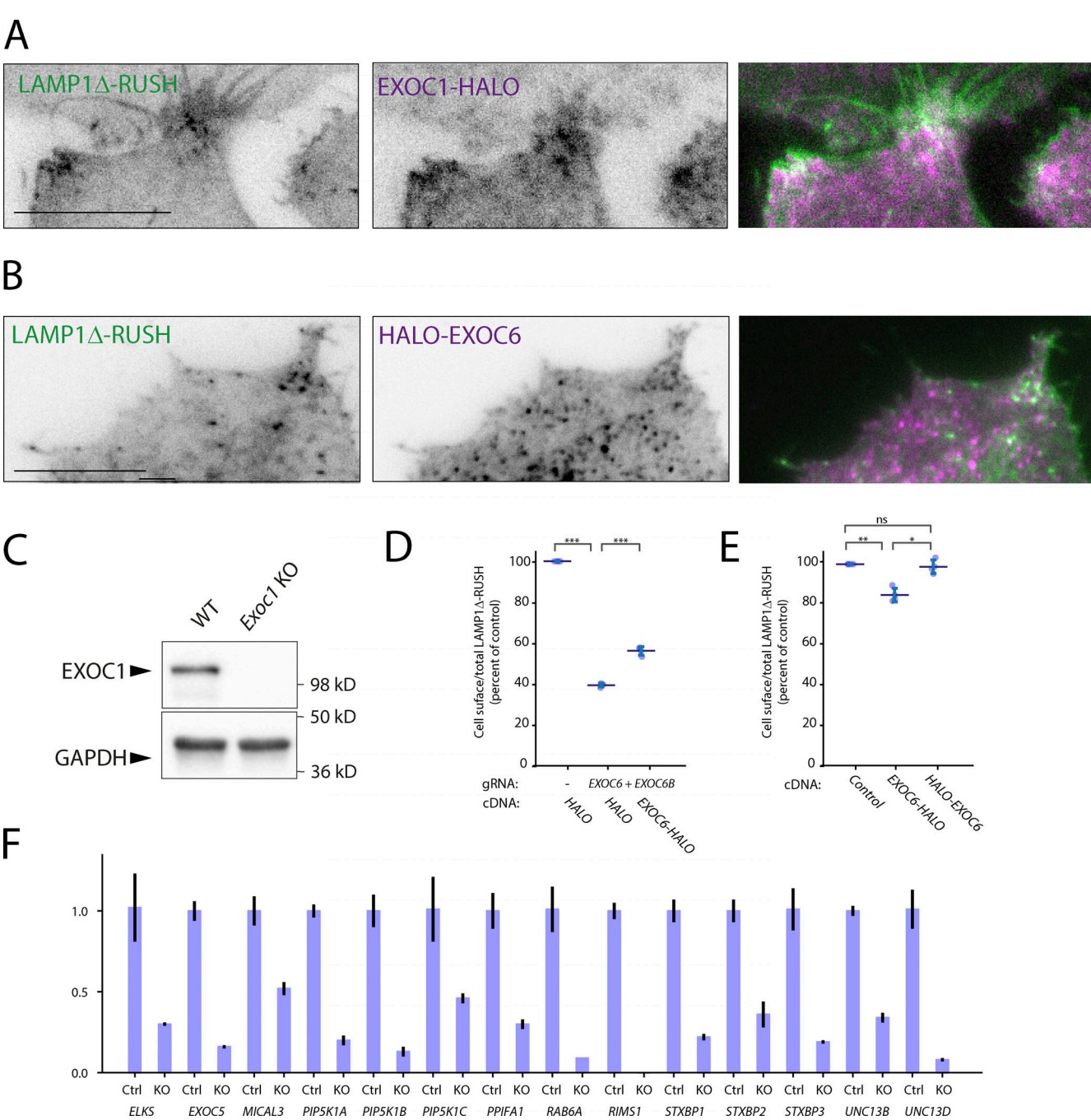

Figure S3.   **Localization of exocyst subunits to LAMP1Δ-RUSH post-Golgi carriers, a C-terminal tag negatively affects EXOC6 function and validation of gene abrogation. (A)** TIRF imaging of LAMP1Δ-RUSH (gray) cells expressing heterologous EXOC1-HALO (purple) after ∼35 min in biotin. From Video 11. Scale bar: 10 µm **(B)** TIRF imaging of LAMP1Δ-RUSH (gray) cells expressing heterologous HALO-EXOC6 (purple) after ∼35 min in biotin. From Video 12. Scale bar: 10 µm. **(C)** Immunoblot confirming loss of EXOC1 in a transient LAMP1Δ-RUSH EXOC1 KO cell population. **(D)** Recovery of EXOC6-HALO to EXOC6+6B KO cells, demonstrating a less efficient recovery than N-terminally tagged EXOC6 (Fig. 3 A). **(E)** Overexpression of EXOC6-HALO has a moderate but significant effect on cell-surface delivery of LAMP1Δ-RUSH. **(F)** Validation of guide RNA KO efficiency by qRT-PCR. Data from all conditions was internally normalized to GAPDH expression and is represented as fold change of control LAMP1Δ-RUSH Cas9 cells. ELKS2/ERC2 and UNC13C were not detectable by qPCR in control conditions. Error bar = SD of at least three independent experimental repeats. Tukey's multiple comparisons test (HSD, FWER = 0.05) was performed on data in D and E. *P ≤ 0.05; **P ≤ 0.01; ***P ≤ 0.001.

GFP-LAMP1Δ-RUSH  LAMP1Δ-GFP-RUSH

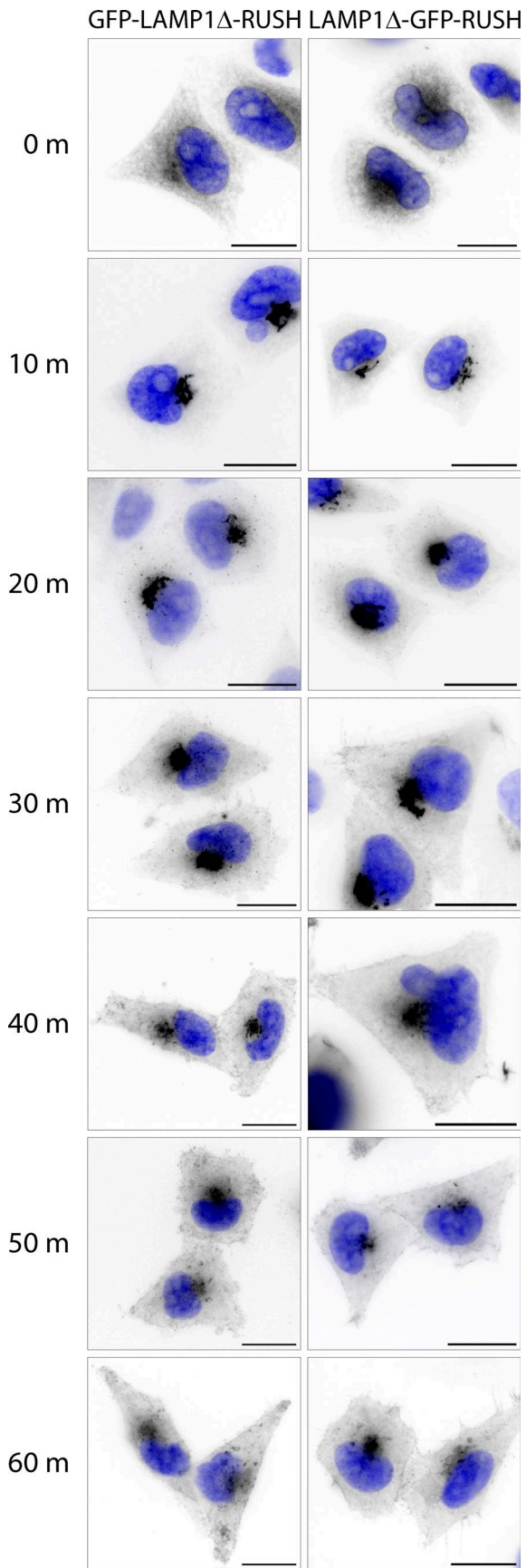

Figure S4.   **C- or N-terminally tagged LAMP1Δ-RUSH traffics with comparable kinetics to the cell surface.** Widefield imaging series across different incubations of biotin (time indicated). Nucleus stain = DAPI. Scale bar: 20 μm. m, min.

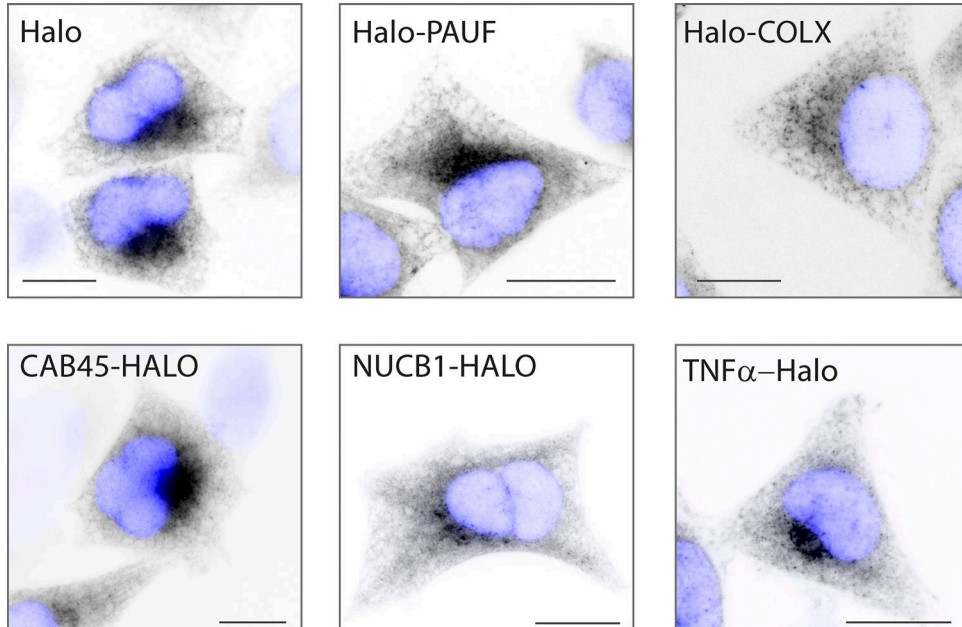

Figure S5. **Different soluble RUSH cargos are effectively retained in the ER.** Widefield imaging of stable cell lines expressing different RUSH cargos. Images show cargos effectively retained in the ER in the absence of biotin. Scale bar: 10 µm.

Video 1. **Lattice-SIM live-cell imaging of stable LAMP1Δ-RUSH HeLa cells showing the type-1 membrane spanning RUSH reporter LAMP1Δ-GFP trafficking from the ER (0′) to the Golgi apparatus (15′) and then directly to the plasma membrane (30′–65′).** Cells were imaged every 30 s for 1 h after addition of biotin (500 µM). Scale bar: 10 µm. 10 fps. See Fig. 1 B for stills.

Video 2. **Lattice-SIM live-cell imaging of LAMP1Δ-RUSH leaving the Golgi apparatus of HeLa cells in tubular carriers 35 min after biotin addition (500 µM).** Cells were imaged every 1.6 s for 5 min. Scale bar: 10 µm. 10 fps. See Fig. 1 C for stills.

Video 3. **Lattice-SIM live-cell imaging of a HeLa cell stably expressing LAMP1Δ-RUSH and transfected with β-tubulin-mCherry (24 h).** Video depicts LAMP1Δ-RUSH carriers (green) moving along the microtubular network (magenta). The cell was imaged 34 min after addition of biotin (500 µM) every 1.6 s for 2.4 min. Scale bar: 10 µm. 10 fps. See Fig. 1 D for stills.

Video 4. **TIRF live-cell imaging of stable LAMP1Δ-RUSH HeLa cells 35 min after biotin addition (500 µM).** Video shows the plasma membrane TIRF plane where LAMP1Δ-RUSH post-Golgi tubules can be seen fusing. Cells were imaged every 30 ms for 7 s. Scale bar: 10 µm. 50 fps. See Fig. 1 E for stills.

Video 5. **Lattice-SIM live-cell imaging of a HeLa cell overexpressing HALO-RAB6A (24 h).** Heterologous RAB6A is present at the Golgi and in tubular structures that bud off and travel toward the plasma membrane. The cell was imaged every 3.19 s for 4 min. Scale bar: 10 µm. 20 fps. See Fig. 2 A for stills.

Video 6. **Lattice-SIM live-cell imaging of a HeLa cell stably expressing LAMP1Δ-RUSH and transfected with HALO-RAB6A (24 h).** Video shows HALO-RAB6A (magenta) co-localizing with LAMP1Δ-RUSH tubules (green) leaving the Golgi and moving toward the plasma membrane. The cell was imaged 21 min after biotin addition (500 µM), every 3.16 s for 4.2 min. Scale bar: 10 µm. 10 fps. See Fig. 2 B for stills.

Video 7. **Lattice-SIM live-cell imaging of a HeLa cell stably expressing LAMP1Δ-RUSH and transfected with HALO-ARHGEF10 (24 h).** Video shows co-localization of HALO-ARHGEF10 (magenta) with LAMP1Δ-RUSH carriers (green) traveling toward the plasma membrane. The cell was imaged 32 min after biotin addition (500 µM), every 3.16 s for 7.2 min. Scale bar: 10 µm. 10 fps. See Fig. 2 D for stills.

Video 8. **TIRF live-cell imaging of a HeLa cell stably expressing LAMP1Δ-RUSH and transfected with HALO-RAB8A (24 h).** Video shows plasma membrane TIRF microscopy plane, where HALO-RAB8A (magenta) can be seen co-localizing with LAMP1Δ-RUSH carriers (green) near their fusion site. The cell was imaged 18 min after biotin every addition (500 µM), every 100 ms for 5.12 min. Scale bar: 10 µm. 50 fps. See Fig. 2 F for stills.

Video 9. **TIRF live-cell imaging of a EXOC3-KO HeLa cell stably expressing LAMP1Δ-RUSH and transfected with HALO-EXOC3 (24 h).** Video shows co-localization of EXOC3-HALO (magenta) with LAMP1Δ-RUSH (green) carriers. The cell was imaged 17 min after biotin addition (500 µM), every 110 ms for 6 min. Scale bar: 10 µm. 50 fps. See Fig. 4 A for stills.

Video 10. **TIRF live-cell imaging of a HeLa cell stably expressing LAMP1Δ-RUSH and transfected with HALO-ELKS (24 h).** Video shows LAMP1Δ-RUSH carriers (green) fusing in plasma membrane sites enriched for HALO-ELKS (magenta). Cells were imaged 31 min after biotin addition (500 µM), every 110 ms for 3.4 min. Scale bar: 10 µm. 100 fps. See Fig. S2 A for stills.

Video 11. **TIRF live-cell imaging of a HeLa cell stably expressing LAMP1Δ-RUSH and transfected with EXOC1-HALO (24 h).** Video shows LAMP1Δ-RUSH carriers (green) co-localizing with EXOC1-HALO (magenta) near their fusion site. Cells were imaged 40 min after biotin addition (500 µM), every 50 ms for 3.15 min. Scale bar: 10 µm. 100 fps. See Fig. S3 A for stills.

Video 12. **TIRF live-cell imaging of a HeLa cell stably expressing LAMP1Δ-RUSH and transfected with HALO-EXOC6 (24 h).** Video shows LAMP1Δ-RUSH carriers (green) co-localizing with HALO-EXOC6 (magenta) near their fusion site. Cells were imaged 17 min after biotin addition (500 µM), every 110 ms for 6.2 min. Scale bar: 10 µm. 50 fps. See Fig. S3 B for stills.

**Provided online are three tables. Table S1 lists gRNA sequences used for this work. Table S2 details the post-analysis secretomics data set used in this study. Table S3 lists all oligos used for qPCR.**

