## [Peer Review File · The Journal of Cell Biology]

The exocyst complex is an essential component of the mammalian constitutive secretory pathway

Conceição Pereira, Danièle Stalder, Georgina Anderson, Amber Shun-Shion, Jack Houghton, Robin Antrobus, Michael Chapman, Daniel Fazakerley, and David Gershlick

Corresponding Author(s): David Gershlick, University of Cambridge

Review Timeline:

Submission Date:	2022-05-31
Editorial Decision:	2022-07-13
Revision Received:	2022-11-11
Editorial Decision:	2022-12-14
Revision Received:	2023-01-26

Monitoring Editor: Ian Macara

Scientific Editor: Tim Fessenden

Transaction Report:

DOI: <https://doi.org/10.1083/jcb.202205137>

July 13, 2022

Re: JCB manuscript #202205137

Dr. David C. Gershlick
University of Cambridge
Cambridge Biomedical Campus
CB2 0XY
United Kingdom

Dear David -

We have now received feedback about your interesting manuscript "The exocyst complex is an essential component of the mammalian constitutive secretory pathway" from three external reviewers with expertise in the field. As you will see from their appended comments, although reviewer #3 is positive, the other two reviewers have substantial issues with the novelty of the findings, the rigor of analyses, and conclusions that are not always supported by the data. Unfortunately, therefore, based on the reviewer comments, we are unable to consider the manuscript for publication in its present form.

To highlight some of the key issues, Reviewer #1 feels that you do not adequately justify the idea that the role of exocyst in soluble protein secretion is "controversial"; they are unclear why a mutant protein that is not secreted is used as the RUSH cargo; and note that the colocalization of cargo with exocyst is not compelling. They comment that many of the results in the manuscript are descriptive rather than robustly quantitative, and feel that glycosylation as a marker of arrival at the PM is not definitive. They also wonder how you know when vesicles fuse, in the dual color movies (Video 3A), and they do not consider a ~20% drop in Lamp exocytosis in response to RalA/B knockout to demonstrate essentiality. Finally, they question the use of the specific (and sometimes different) exocyst subunits in the study.

Reviewer #2 also seems confused about the choice of exocyst subunits for the experiments, and the fact that different ones are used in different assays. There are also issues about their over-expression and whether this recapitulates the behavior of the endogenous subunits. This reviewer also questions the Ral data, and feels that the novelty of the last two figures is unclear as exocyst is well known to play a major role in delivery of cargo to the PM.

We feel that the comments of these reviewers are justified and would need substantial additional experimental data to be addressed satisfactorily. For this reason you may elect to send the manuscript elsewhere. If you wish to submit a revised manuscript to JCB, we will need a point-by-point response to each of the reviewer comments, with additional data as requested, and the manuscript will have to be re-evaluated by reviewers # 1 and 2.

Please let us know if you are able to address the major issues outlined above and wish to submit a revised manuscript to JCB. Note that a substantial amount of additional experimental data likely would be needed to satisfactorily address the concerns of the reviewers. The typical timeframe for revisions is three to four months. While most universities and institutes have reopened labs and allowed researchers to begin working at nearly pre-pandemic levels, we at JCB realize that the lingering effects of the COVID-19 pandemic may still be impacting some aspects of your work, including the acquisition of equipment and reagents. Therefore, if you anticipate any difficulties in meeting this aforementioned revision time limit, please contact us and we can work with you to find an appropriate time frame for resubmission. Please note that papers are generally considered through only one revision cycle, so any revised manuscript will likely be either accepted or rejected.

If you choose to revise and resubmit your manuscript, please also attend to the following editorial points. Please direct any editorial questions to the journal office.

GENERAL GUIDELINES:

Text limits: Character count is < 40,000, not including spaces. Count includes title page, abstract, introduction, results, discussion, and acknowledgments. Count does not include materials and methods, figure legends, references, tables, or supplemental legends.

Figures: Your manuscript may have up to 10 main text figures. To avoid delays in production, figures must be prepared according to the policies outlined in our Instructions to Authors, under Data Presentation, <https://jcb.rupress.org/site/misc/ifora.xhtml>. All figures in accepted manuscripts will be screened prior to publication.

*****IMPORTANT:** It is JCB policy that if requested, original data images must be made available. Failure to provide original images upon request will result in unavoidable delays in publication. Please ensure that you have access to all original microscopy and blot data images before submitting your revision. ***

Supplemental information: There are strict limits on the allowable amount of supplemental data. Your manuscript may have up to 5 supplemental figures. Up to 10 supplemental videos or flash animations are allowed. A summary of all supplemental material should appear at the end of the Materials and methods section.

Please note that JCB now requires authors to submit Source Data used to generate figures containing gels and Western blots with all revised manuscripts. This Source Data consists of fully uncropped and unprocessed images for each gel/blot displayed in the main and supplemental figures. Since your paper includes cropped gel and/or blot images, please be sure to provide one Source Data file for each figure that contains gels and/or blots along with your revised manuscript files. File names for Source Data figures should be alphanumeric without any spaces or special characters (i.e., SourceDataF#, where F# refers to the associated main figure number or SourceDataFS# for those associated with Supplementary figures). The lanes of the gels/blots should be labeled as they are in the associated figure, the place where cropping was applied should be marked (with a box), and molecular weight/size standards should be labeled wherever possible. Source Data files will be made available to reviewers during evaluation of revised manuscripts and, if your paper is eventually published in JCB, the files will be directly linked to specific figures in the published article.

If you choose to resubmit, please include a cover letter addressing the reviewers' comments point by point. Please also highlight all changes in the text of the manuscript.

Regardless of how you choose to proceed, we hope that the comments below will prove constructive as your work progresses. We would be happy to discuss them further once you've had a chance to consider the points raised. You can contact the journal office with any questions, cellbio@rockefeller.edu or call (212) 327-8588.

Thank you for thinking of JCB as an appropriate place to publish your work.

Sincerely,

Ian Macara, Ph.D.
Editor

Andrea L. Marat, Ph.D.
Senior Scientific Editor

Journal of Cell Biology

Reviewer #1 (Comments to the Authors (Required)):

The manuscript submitted to JCB entitled "The exocyst complex is an essential component of the mammalian constitutive secretory pathway" by the Gershlick laboratory uses the 'RUSH' hook system to examine delivery of a modified LAMP1 cargo (and other cargo) to the plasma membrane. Specifically, they examine the effect of exocyst knockout (KO) on the presumed plasma membrane delivery and claim a specific exocyst block on general secretion of both a membrane protein (LAMP) and secreted cargo. In principle this study could be useful and the finding that soluble secreted proteins might be affected is intriguing, but there are serious limitations in both the novelty of the findings, depth and rigor of analysis, and consideration of alternatives/controls.

Main Critiques:

1. Stated Premise/Novelty:

The stated motivation was the "controversial" role of the exocyst complex in soluble protein secretion in which exocyst antibodies had no effect, but EXOC7/Exo70 knockdown decreased VSVG delivery. They stated "Although exocyst is essential for endosomal recycling to the plasma membrane^{26,29}, the role of exocyst in biosynthetic protein secretion remains unclear. Inhibition of exocyst with antibodies does not affect delivery of tsVSV-G, a marker of the secretory pathway, to the plasma membrane^{23,27}, however, depletion of EXOC7 decreases tsVSV-G delivery to the plasma membrane³⁰. It is therefore not known and remains controversial if exocyst plays a direct role in soluble protein secretion in mammalian cells^{2,26}." There are several issues.

- a. VSVG is a membrane protein so any mixed evidence here does NOT imply at what the outcome would be for soluble protein secretion (here they would need to look at NPY or other soluble reporters).
- b. The lack of an effect by Antibodies could trivially be due to the state of the exocyst (see this 2016 paper - PMID: 27376061)

c. (minor) They mention known effects on recycling pathways, but cite general reviews and not the primary articles, which they should do.

2. RUSH Approach. The authors draw most conclusions using this system, which is used to pulse a wave of exocytosis. It is however not clear if the findings here are the same or different with other published approaches using temperature sensitive mutants (VSVGts045) or FM-tags. More confusing the probe that they use is not a traditional PM targeted protein, but one based on LAMP which had its endocytic moiety blocked. (Why did they pick this?) Also unlike the other work on VSVG that KD EXOC7/Exo70 the authors chose to EXOC3/Sec6 and EXOC5/Sec10. Why? What is there rationale. Here it would make sense to test VSVGts045 with EXOC3 and EXOC5 to confirm the other results. But even here the KO (or KD) is chronic so it's hard to know if the KD or KO effect is direct or indirect - a topic which should be addressed.

3. Colocalization. Here it was claimed in the abstract and elsewhere that "exocyst co-localized with carriers fusing at the plasma membrane" This is not compelling. The movies do not show much colocalization. The fusion in these dual colour movies (Video 3A) is not clear - how do they know in this movie when vesicles fuse? This has not been rigorously addressed. Same for the cursory claim of fusion hotspots. Also why are they imaging with EXOC1 (Sec3-Halo) when they show a functional role with EXOC3 and EXOC5 (which should be used)?! (They should use EXOC3 or 5). The data in Fig. 3 are NOT compelling and if anything, the movie (Video 3A) shows many LAMP vesicles without exocyst.

4. RALA/B. This is claimed as "essential" but loss of both led to only a 20% drop of Lamp exocytosis. This does not appear to be essential.

5. Robustness of findings. Many findings are presented in a descriptive rather than a robust quantitative manner including cargo, fusion, hotspots, etc. The authors say "this [LAMP1] fusion have comparable kinetics". But no fusion analysis is shown? How long does vesicle fusion take? Maybe true but no analysis is shown.

6. Readout for plasma membrane arrival. They claim of glycosylation of LAMP is used as an indicator of arrival at the PM may not be correct. It is possible that the glycosylation seen happens at the Golgi. To support their point they should show that BFA or 20 degree TGN blocks the glycosylation. If glycosylation still occurs after BFA then the readout would not follow a Golgi to plasma membrane step. If so their assay or interpretation would need to be adjusted.

7. Fig. 5. The key data is from "C", but what is analyzed, the change in glycosylation? See above but how do they know the block was earlier than fusion? Also in Fig. 5D the authors should quantitate the levels of cargo inside the cells in +/- EXOC3 to see how this compares with the glycosylation.

8. Nomenclature/Approach. Exocyst is being used as a term when mainly EXOC3 is examined. They assume that all subunits have similar roles, but several papers (uncited) show that Sec10 had a differential role in MDCK cells. Really other subunits should be examined and the cells should be rescued, ideally in an acute manner using perhaps release of a mitochondrial targeted exocyst subunit. This is a heavy lift, but the authors are not circumspect in the limitations of their approach and in context relative to other prior work.

9. Functional effect? Which stage of secretion is potentially blocked in the KO? Tethering? Fusion? This would seem to be an important element to characterize.

Minor:

- Video 1E looks encouraging, but why is the movie for only 7 seconds shown? Why not show the full movie (e.g. several minutes?) as done elsewhere?
- NPY. This and other soluble probes should be used to see if they use the exocyst? If they do not, how do they rationalize it especially as this cargo was reported to use ELKS as a tether (?)

Overall there are some intriguing findings and if it was true that most soluble cargo require the exocyst to secrete than this would be important. However, to make this claim really requires a high burden of proof - which for the reasons noted above this manuscript has not hit the level of rigor needed for JCB.

Reviewer #2 (Comments to the Authors (Required)):

In this manuscript, the authors use a variety of experimental approaches to study the sorting route from the Golgi apparatus to the plasma membrane. Suppression of core exocyst subunits reduced carrier fusion and caused cargo accumulation in the post-Golgi carriers. Overall, the authors make a case that the exocyst complex is a critical component of the mammalian constitutive secretory pathway. The findings are interesting. The work would benefit by addressing several issues.

General comments:

1. A brief literature summary of some noncanonical functions of the exocyst such as innate immune function and autophagy would be relevant in the discussion. Perhaps the constitutive and homeostatic nature of these secretory vesicles plays a specific role in cell biologically relevant contexts. Discussing this in a paragraph would improve the manuscript from a cell biological context.

Specific comments:

1. "After synthesis in the ER, LAMP1 traffics via the Golgi apparatus to the plasma membrane where it is endocytosed, in clathrin-coated vesicles, to be delivered to the endolysosomal system and finally to the lysosome"
The authors focus on the claim that defective LAMP1 accumulates on the plasma membrane. However, direct accumulation of LAMP1 on early/sorting endosome has previously been reported. Perhaps the defective LAMP1 has only a membrane directed route available for these structures. The authors should nevertheless clarify this.
2. EXOC1-HALO experiment: The authors do not mention why EXOC1 (Sec3) was selected for in this analysis. Generally, in mammalian cells EXOC3 (Sec6) and EXOC4 (Sec8) (Exocyst is also called Sec6/8 complex), are used as a proxy for the holocomplex, since several subcomplexes of the exocyst have been reported. Specifically, secretory vesicle fusion by the exocyst requires two distinct complexes coming together and only EXOC3 and EXOC4 have been shown to be part of both of them, where EXOC3 is not. The authors need to clarify.
3. Notably, EXOC4/Sec8-RFP based TIRF microscopy tools have been used previously and well established in the field. EXOC4 modifications were shown to not affect holocomplex formation as well, which is not the case for EXOC1. The authors should justify the selection of EXOC1 in this regard. Does heterologous EXOC1 localize in the holocomplex? The authors should show this.
Also why did the authors switch to EXOC6 (Sec15) for the subsequent assay? Consistency in prioritizing the subunits for the assay is crucial since their participation in subcomplexes can vary. The authors should explain and preferably perform experiments solely with canonical EXOC3/Sec6 and EXOC4/Sec8 for both of these assays.
4. Carrier IP: Authors should also show EXOC2 (Sec5) and EXOC8 (Exo84). These are two of the most important members of the exocyst subcomplexes reported.
5. The authors should show validation for the RALA and RALB KO validation, since RALGTPases, (RALA and RALB) are the major, if not, the prime regulators of the exocyst in mammalian cells validated by prior literature. PIP5K data is interesting, but lacks depth as a stimulatory node upstream. PIP5K perhaps is mediating docking of the exocyst complex, not formation of the holocomplex. If otherwise, the authors should delineate how PIP5K can drive exocyst holocomplex assembly at the tethering site.
6. The exocyst's functional role as a major machinery for secretion is well understood. This function has been reported to be non-specific and constitutive. The novelty of last two figures in this regard is unclear.

Reviewer #3 (Comments to the Authors (Required)):

In this manuscript Pereira et al. use a combination of trafficking assays, imaging and transient CRISPR-Cas9 knock out to test the involvement of the exocyst in constitutive secretion. Using a LAMP1 mutant lacking the endocytic trafficking motif as a reporter for quantitative RUSH assays they show that secretory carriers traffic on microtubule tracks and are positive for the small GTPase Rab6 and Rab8 and ARHGEF10. KO of Rab6 causes accumulation of the reporter in the Golgi. Further employing their assays, they show that the exocyst subunit EXOC1 localized to fusion hot spots together with the secretory carriers and KO of two exocyst subunits EXOC3 and 5 causes accumulation of the LAMP1 reporter in the cell periphery. Interesting when KO the other plasma membrane tether ELKS, secretion of the reporter is not affected. Taking advantage of a cytoplasmic tag on the reporter they are able to purify the transport carriers and show enrichment of exocyst subunits and the small GTPase Rab6. KO of PIP5Ks and RalA/B affected secretion of the reporter. The authors then nicely show that EXOC3 KO disrupt secretion of various known secretory markers. Employing mass spectrometry to study the secretome of EXOC3 KO cells they show that many soluble cargoes cannot reach the cell surface. KO of EXOC3 in adipocytes and lymphocytes blocks the secretion of physiological cargoes, strengthening the message of the paper.

Overall, this manuscript fills an important gap in the knowledge of the function of the exocyst and brings solid evidence of its involvement in direct secretory trafficking from the Golgi to the plasma membrane using solid kinetic assays and also providing specific disruption of the endogenous secretome. The paper is very nicely written and easy to read, experiments are well executed and presented. Therefore, I would recommend acceptance of the manuscript with the following minor revisions:

- Figure 2: color coding in the crops is a bit confusing. I understand that the authors want to use the colors of the labels to frame the crop so shouldn't the reporter crop green? It is cyan at the moment.
- Figure 2: Arrows to highlight the co-localizing structures would help.
- Supplementary Figure 1: hard to see the co-localization with this LUT and without single color images/crops.
- Interesting that ELKS localizes to the same hot spots as EXO and the carriers but KO of ELKS does not affect secretion of the reporters at all? Is this worth mentioning in the discussion?
- Figure 3A: It would be useful to have a crop to better show the co-localization. Hard to see with a low quality overview image.
- Figure 4A: Aren't figure 3A and 4A showing the same thing with two different exocyst subunits? Again crops and arrows would be useful.
- Figure 4D-E While the effect of PIP5K is obvious the effect of the double RalA/B KO is only modest. The authors should carefully rephrase their conclusions.
- Line 172: I cannot find any supplementary figure 4.
- Line 172: "We then performed RUSH assays and used an unbiased FACS assay to quantitatively monitor the loss of these

soluble cargoes". Does it mean that quantification in figure 5C was carried out in a different way as explained in 1G? The authors could explain in the figure legend? Are all RUSH assays quantified using FACS as explained in the methods?

- In general: bars in graphs are SD? SE?

UNIVERSITY OF
CAMBRIDGE

Dr. David Gershlick PhD
Sir Henry Dale Fellow
Cambridge Institute for Medical Research
University of Cambridge

11th November 2022

Dear Reviewers,

We are pleased to resubmit our manuscript entitled "*The exocyst complex is an essential component of the mammalian constitutive secretory pathway*".

We thank the Reviewers for their comments on our manuscript. We are grateful for the in-depth reviews, and we have addressed them by providing substantial additional data amounting to 17 new data panels. We have summarised these data and clarifications in a point-by-point response below. We believe these additions have improved the manuscript and strengthened our findings.

We have performed the following:

- Extensive quantification of colocalisations for RAB6a, ARHGEF10, RAB8a, EXOC1, EXOC6 with post-Golgi carries.
- Live-cell structured-illumination imaging demonstrating the colocalisation of the carriers with RAB6A.
- Novel TIRF imaging of EXOC3-HALO as well as quantification of its colocalisation with post-Golgi carriers.
- KO and recovery of all 8 exocyst subunits, demonstrating the necessity of any of the 8 subunits for post-Golgi carrier delivery to the cell surface.
- Corresponding quantification and imaging of all 8 KO conditions showing consistent phenotype in all conditions.
- Novel and important data comparing the effect of overexpressing EXOC6-HALO vs HALO-EXOC6 in mammalian cells, demonstrating that EXOC6-HALO is functionally abrogated in comparison to HALO-EXOC6.
- Novel biochemical purification of post-Golgi secretory carriers with all 8 canonical exocyst subunits demonstrating that all subunits are recruited to post-Golgi carriers in an unbiased fashion, orthogonal to and in support of our imaging data.
- A new imaging panel on the phenotype of PIP5K KO reveals a post-Golgi phenotype comparable to the loss of exocyst subunits.
- An imaging panel experimentally validating comparable kinetics of LAMP1-GFP cells vs GFP-LAMP1 cells.
- Novel EXOC3-KO data in adipocytes demonstrating a significant reduction of secretion of the hormone leptin.

These experiments are also detailed in the specific response to the Reviewers below. We have also revised the text and figures for clarity, and to incorporate the new changes.

The editor and the Reviewers highlighted some common themes that we wish to address here.

With regard to the novelty of our findings. There is no published evidence, to our knowledge, that exocyst plays a role in the constitutive secretion of soluble proteins in mammalian cells, and there is limited data on the role of exocyst in biosynthetic membrane protein sorting. There is evidence from yeast studies; however, in mammalian cells, the contrary is true, and ELKS has been implicated as the tether in mammalian cells for constitutive secretory cargo (PMIDs: 32521280, 31142554). The evidence for exocyst's role in mammalian cells has been reserved for major roles in endosomal recycling to the cell surface, targeted protein delivery in polarised mammalian cells, ciliogenesis and cytokinesis, as well as a role in innate immunity and autophagy, which are well evidenced in the literature (cited in the revised manuscript). Constitutive secretion is of absolute importance to cell biology processes, including the secretion of signalling molecules (e.g. wnts), hormones (e.g., leptin), antibodies and collagen (which accounts for 30% of mammalian protein by mass). As Reviewer one states, "if it was true that most soluble cargo require the exocyst to secrete than this would be important", and Reviewer three states, "this manuscript fills an important gap in the knowledge of the function of the exocyst and brings solid evidence of its involvement in direct secretory trafficking from the Golgi to the plasma membrane using solid kinetic assays and also providing specific disruption of the endogenous secretome".

Regarding the use of glycosylation to detect the cell surface arrival of cargo. We apologise for this confusion and clarify that glycosylation was not used to detect the cell surface arrival of the cargo. The panel in figure one, demonstrating the glycosylation, was only to show that the cargo trafficks through the Golgi apparatus,

where the majority of the glycosylation machinery resides, and thus uses a canonical route through the secretory pathway of the cell. The quantification below the glycosylation in the original figure 1 is based on our cell surface flow cytometry assay that we developed for this study, as outlined in the original manuscript "To quantitatively study this secretory route, we have developed FACS-based quantitative cell-surface protein delivery assay (Fig.1G). By using a purified anti-GFP nanobody fused to a fluorescent mCherry, we can quantify delivery to the plasma membrane of the integral membrane protein cargo. (Fig. 1H)". We have clarified this in several ways in this resubmission:1) clarified the text, 2) moved the glycosylation data, which demonstrates Golgi passage and proper processing to the supplement, 3) revised the figure legend. Finally, we wish to stress the level of quantitation and statistical rigour of this assay we have developed for this study, where each experiment involves individually ratiometrically measuring a minimum of 30,000 cells, and thus (taking account of biological repeats) each experiment is a measurement of a minimum of 90,000 cells. This is the same assay we used in Figures 2H, 3A, 4G, S2B,D,E,F and S3D,E.

The final common theme was our choice of exocyst subunits used for analysis. To address this, we have performed multiple new experiments to allow for a more comprehensive and cohesive manuscript. We now demonstrate that loss of any of the 8 canonical exocyst subunits results in defective cell surface delivery from the Golgi apparatus (new Fig. 3A) and have added a new figure demonstrating that the point of defect is post-Golgi export and before cell surface fusion (new Fig. 3B). We have repeated our novel 'carrier-IP' biochemical analysis of the recruitment of exocyst to post-Golgi carriers and extended it to all canonical exocyst components, an unbiased approach demonstrating that all members of the complex are recruited to the carriers (new Fig. 4F). We have selected EXOC3 as a representative for the complex as suggested by Reviewers one and two and demonstrate that EXOC3 is recruited to carriers (new Fig. 4A), with both statistically analysed imaging (new Fig. 4B) and unbiased biochemistry (new Fig. 4F). Loss of EXOC3 affects the secretion of multiple soluble cargoes (new Fig. 5C,D), as well as endogenous soluble cargoes by secretomics (new Fig. 6A,B) and in physiological cell types (new Fig. 6C-H). For all colocalisation experiments, we have now performed extensive quantification and statistics (new Fig. 2C,E,G, Fig. 4B,C,D). Together these new experiments and statistical analyses support our findings and strengthen the manuscript; we thank the Reviewers for this.

We have addressed specific concerns by Reviewers in a point-by-point manner below.

Reviewer 1:

The manuscript submitted to JCB entitled "The exocyst complex is an essential component of the mammalian constitutive secretory pathway" by the Gershlick laboratory uses the 'RUSH' hook system to examine delivery of a modified LAMP1 cargo (and other cargo) to the plasma membrane. Specifically, they examine the effect of exocyst knockout (KO) on the presumed plasma membrane delivery and claim a specific exocyst block on general secretion of both a membrane protein (LAMP) and secreted cargo. In principle this study could be useful and the finding that soluble secreted proteins might be affected is intriguing, but there are serious limitations in both the novelty of the findings, depth and rigor of analysis, and consideration of alternatives/controls.

We thank the Reviewer for their comments. As detailed below, we have revised the manuscript with additional text and experiments that address these points.

Main Critiques:

1. Stated Premise/Novelty:

The stated motivation was the "controversial" role of the exocyst complex in soluble protein secretion in which exocyst antibodies had no effect, but EXOC7/Exo70 knockdown decreased VSVG delivery. They stated "Although exocyst is essential for endosomal recycling to the plasma membrane^{26,29}, the role of exocyst in biosynthetic protein secretion remains unclear. Inhibition of exocyst with antibodies does not affect delivery of tsVSV-G, a marker of the secretory pathway, to the plasma membrane^{23,27}, however, depletion of EXOC7 decreases tsVSV-G delivery to the plasma membrane³⁰. It is therefore not known and remains controversial if exocyst plays a direct role in soluble protein secretion in mammalian cells^{2,26}." There are several issues.

a. VSVG is a membrane protein so any mixed evidence here does NOT imply at what the outcome would be for soluble protein secretion (here they would need to look at NPY or other soluble reporters).

We agree with the Reviewer that the nature of VSV-G as a membrane protein does not demonstrate exocyst has a role in soluble protein secretion. We, therefore, agree the role of exocyst has not been studied in the

context of soluble protein secretion in mammalian cells. This was a point we were attempting to make to provide a rationale for our study, and we have clarified this point in the revision. We agree that in order to study the role of exocyst in soluble protein secretion this needs to be studied directly. Although the Reviewer suggests NPY, which is soluble, this is not an appropriate protein to study in this context as it undergoes regulated secretion, not constitutive - a different process and thus outside the scope of this study.

Manuscript figures 5 and 6 in the original study directly address the constitutive secretion of soluble proteins. We demonstrate in these figures that exocyst is important for the secretion of a wide variety of cargoes (Cab45, NUCB1, Collagen, PAUF, and bulk flow secretion by monitoring untagged HALO). Additionally, in figure 6, we demonstrate this in the endogenous context with secretomics demonstrating a drastic decrease in the secretion of soluble proteins. Finally, we observe an exocyst dependence on endogenous constitutively secreted proteins in specialised cell types that secrete antibodies and hormones.

b. The lack of an effect by Antibodies could trivially be due to the state of the exocyst (see this 2016 paper - PMID: 27376061)

We agree with this potential interpretation of the approach taken in the 2001 paper. Thus, we agree that the role of exocyst in the biosynthetic membrane protein sorting to the cell surface in mammalian cells is still of interest. Our manuscript addresses this by overcoming some of the limitations of previous work by studying biosynthetic membrane protein trafficking in the context of exocyst abrogation using transient CRISPR-KO, validated with qPCR, and rescue experiments using exogenous cDNA overexpression. As described above, we have edited our introduction of previous work to provide a clearer perspective and stronger rationale for our study. We have also added this point to the manuscript and cited the 2016 manuscript suggested by the Reviewer.

The updated section reads as the following (page 2, lines 62-68):

*“Inhibition of exocyst with antibodies does not affect delivery of *tsVSV-G*, a marker of the secretory pathway, to the plasma membrane^{23,27}, however, this could be due to ineffective inhibition by antibodies as the epitope may not be exposed under certain exocyst structural conformations⁴⁴. Conversely, some evidence indicates that exocyst is important for biosynthetic sorting to the plasma membrane and depletion of *EXOC7* decreases *tsVSV-G* delivery to the plasma membrane⁴⁵. It is therefore of interest to examine if exocyst has a direct role in biosynthetic membrane protein sorting in mammalian cells. Moreover, it is not known if exocyst is necessary or important for soluble protein secretion in mammalian cells^{2,26}.”*

c. (minor) They mention known effects on recycling pathways, but cite general reviews and not the primary articles, which they should do.

We thank the reviewer for this point. We have addressed this in the revised manuscript.

*2. RUSH Approach. The authors draw most conclusions using this system, which is used to pulse a wave of exocytosis. It is however not clear if the findings here are the same or different with other published approaches using temperature sensitive mutants (*VSVGts045*) or *FM-tags*. More confusing the probe that they use is not a traditional PM targeted protein, but one based on *LAMP* which had its endocytic moiety blocked. (Why did they pick this?)*

To our knowledge, these alternative kinetic trafficking assays have not been performed in the context of exocyst abrogation other than as discussed above and cited in the manuscript (PMIDs: 17761530, 32639540). Additionally, we agree with a generalised critique of all these kinetic trafficking experimental systems: there is not only an overexpression of cargo but also a non-physiological wave of cargo sorting through the secretory pathway.

We chose to use the RUSH system, as it 1) avoids non-physiological temperature shifts as in the VSV-G system, 2) can be adapted to a variety of cargoes (e.g., Figure 5), and 3) the LAMP1-RUSH traffics with kinetics to a steady-state level similar to endogenous cargoes (45m-1h), as opposed to VSV-G and the FM system which although excellent models have slower kinetics (VSV-G takes approx 1h30-2h to reach cell surface).

Additionally, using RUSH, there is no detectable organelle stress (PMID: 22406856), and the kinetics of trafficking are in line with endogenous cargoes (PMIDs: 28978644, 19843282). In the submitted manuscript, our findings using the RUSH system have been validated with endogenous soluble protein secretion (New

Fig. 6A,B), in HeLa cells as well as physiologically relevant cell types (New Fig6C-H), which are dependent on exocyst.

The LAMP1 probe was chosen as it has been experimentally demonstrated to traffic directly to the cell surface via the biosynthetic secretory pathway (PMID: 28978644), and thus can act as an orthologous validated marker of this pathway, which we demonstrate further in Figure 2. LAMP1 has been demonstrated to tolerate a tag on the N and C termini without affecting the trafficking or kinetics (PMID: 28978644); we have also added an imaging panel to this effect (new Fig. S4). In addition, we note that the VSV-G and LAMP1-RUSH have been demonstrated to take the same pathway to the cell surface (28978644). A modified version of this paragraph has been included in the new discussion to help the readership (page 12, lines 245-251):

“We chose LAMP1 Δ YQTI as a probe as it has been experimentally demonstrated to traffic directly to the cell surface via the biosynthetic secretory pathway¹, and thus can act as an orthologous validated marker of this pathway, as demonstrated in Figures 1 and 2. Additionally, LAMP1 can tolerate a tag on the N and C termini without affecting the trafficking or kinetics¹ (Fig. S4). Some studies suggest that LAMP1 traffics via the endolysosomal system to the lysosome⁶² (the so-called ‘direct pathway’), and we cannot rule out a non-detectable subset of the RUSH cargo taking this pathway. Nevertheless, with the LAMP1 Δ YQTI-RUSH we observe a significant amount of trafficking directly to the cell surface (Fig. 1) in line with other studies¹.”

Also unlike the other work on VSVG that KD EXOC7/Exo70 the authors chose to EXOC3/Sec6 and EXOC5/Sec10. Why? What is there rationale. Here it would make sense to test VSVGts045 with EXOC3 and EXOC5 to confirm the other results. But even here the KO (or KD) is chronic so it's hard to know if the KD or KO effect is direct or indirect - a topic which should be addressed.

We thank the Reviewer for their comments regarding the subunit choice. In line with suggestions by this Reviewer and Reviewer two, we have repeated a number of studies using EXOC3 as a common surrogate for the whole complex (New Fig. 3A,B, 4A,B,F, 5C,D, 6A,B,D,F,G,H). Additionally, we have added new data that the whole complex is important for biosynthetic cell surface sorting (New Fig. 3) and demonstrates that all complex members are recruited to the carriers via our carrier-IP method (New Fig. 4F). Loss of any of the 8 (with the note that 6 and 6B are redundant) causes a similar defect in secretion, a direct phenotype which is we conclude is not chronic as it is recoverable with exogenous overexpression with guide resistant cDNAs (New Fig 3A). This new dataset includes EXOC7, which the Reviewer points out was originally used for the VSVG studies. We observe EXOC7 recruited to the carriers by the unbiased carrier IP, and that loss of EXOC7 decreases the cell surface delivery of our test carrier to the cell surface consistent with its involvement with biosynthetic membrane sorting.

Our CRISPR/Cas9 knock-outs are transient (i.e., we do not generate stable knock-out cell lines), and we use recoveries to demonstrate the specificity of the abrogation. However, we agree that knockouts can be hard to interpret due to pleiotropic effects. Therefore, in the original submission, we demonstrated colocalisation of EXOC1 and EXOC6 with the LAMP1-carriers proving that exocyst is present on them; additionally, we developed an unbiased biochemical purification of the carriers (that we termed carrier-IP) that orthogonally demonstrates the recruitment of exocyst to these carriers. In the revised manuscript, we have extended this with additional experiments demonstrating that every member of the canonical exocyst complex (with the proviso that 6 and 6B are redundant) causes a decrease in the cell surface delivery of LAMP1 Δ -RUSH, which can be recovered with exogenous overexpression of corresponding cDNAs (New Fig. 3A), causing cargo to accumulate at the exocytic sites (New Fig. 3B). We have additional microscopy data demonstrating significant colocalisation of post-Golgi carriers with overexpressed EXOC3-HALO, in endogenous EXOC3-KO conditions (New Fig. 4A), which we have quantified (New Fig. 4B), and finally, we have extended our Carrier-IP approach with all canonical exocyst subunits, concluding that all are recruited to post-Golgi carriers. Together, we believe that this data demonstrates that exocyst is not only functionally important for biosynthetic membrane sorting and soluble protein secretion but is also localised to the carriers, as we have demonstrated in multiple quantified and orthogonal ways, supporting a model where exocyst functions directly on post-Golgi to plasma membrane carriers.

3. Colocalization. Here it was claimed in the abstract and elsewhere that “exocyst co-localized with carriers fusing at the plasma membrane” This is not compelling. The movies do not show much colocalization. The fusion in these dual colour movies (Video 3A) is not clear - how do they know in this movie when vesicles fuse? This has not been rigorously addressed. Same for the cursory claim of fusion hotspots. Also why are they imaging with EXOC1 (Sec3-Halo) when they show a functional role with EXOC3 and EXOC5 (which should be used)?! (They should use EXOC3 or 5). The data in Fig. 3 are NOT compelling and if anything, the movie (Video 3A) shows many LAMP vesicles without exocyst.

We thank the Reviewer for these comments. To clarify, we conclude simply that the exocyst co-localises prior to fusion as the eventual fate of these carriers is fusion to the cell surface, as demonstrated in figure one of the paper. The fusion hotspots that ELKS localises to are a phenomenon that we reproduce from previous work (PMID: 31142554), and we note that exocyst localises to the same part of the plasma membrane, specifically to the carriers.

To ensure that the claims we make of colocalisation are supported by robust data, we have now, under the Reviewers' advice, undergone an extensive colocalisation analysis followed by statistical testing. From this, we can demonstrate that exocyst subunits are significantly co-localised with post-Golgi LAMP1 positive carriers (New Fig. 2C,E,G, 4B,C,D). We have also performed additional co-localisations of tagged EXOC3 in EXOC3-KO conditions with the carriers (New Fig. 4A), as suggested by this Reviewer and Reviewer two, and followed that up with statistical analysis (New Fig. 4B). In addition, we note that this data is backed up with orthologous data from our carrier-IP approach (New Fig. 4F).

4. RALA/B. This is claimed as "essential" but loss of both led to only a 20% drop of Lamp exocytosis. This does not appear to be essential.

We agree with this point that the loss of both RALs is not as severe a phenotype as the loss of other essential components in this study. This could be due to undiscovered redundancy, the ability of the remaining protein pool, despite 95%+ loss of RNA levels, to compensate, or another mechanism. As our manuscript does not concern the role of the Rals, we have removed this from the updated submission, this has helped with the clarity of the manuscript for the readership, and we thank the Reviewer for the suggestion.

5. Robustness of findings. Many findings are presented in a descriptive rather than a robust quantitative manner including cargo, fusion, hotspots, etc. The authors say "this [LAMP1] fusion have comparable kinetics". But no fusion analysis is shown? How long does vesicle fusion take? Maybe true but no analysis is shown.

We thank the Reviewer for this comment. We would like to note that we considered the manuscript submitted highly quantitative, with each FACS cell-surface RUSH assay experiment comprising a minimum of 90k individual cells per sample, novel unbiased biochemical purification of carriers to demonstrate binding of exocyst subunits to the carriers, a quantitative FACS assay also with 90k minimum cells per sample to demonstrate accumulation of cargo inside the cell, and unbiased statistically analysed proteomics to study the secretome. Across the original submission, excluding the cells used in the biochemical experiments, ELISA, western blots and qPCRs, more than 5 million individual cells were analysed, quantified and statistically evaluated. For all key findings in the original manuscript, data was quantified or an orthogonal unbiased method was used to validate a finding.

Thanks to the use of a kinetic trafficking system where the trafficking route is ER->Golgi->PM, we can make deductions on where a defect is occurring, i.e., prior to plasma membrane fusion. To clarify, when we state, "This fusion had comparable kinetics to the luminal/extracellular tagged", we are referring to the genetic fusion on the opposing termini of the protein, not the fusion of the carriers to the cell surface. We have now clarified this in the original manuscript and added an extra supplemental figure demonstrating the trafficking kinetics of LAMP1 are comparable if tagged N or C terminally (New Fig. S4).

We agree with the Reviewer, however, that there were a number of unquantified claims in the original manuscript. We have since performed a number of additional analyses to increase the rigour of the data. These include quantifying colocalizations for RAB6a (New Fig. 2C), ARHGEF10 (New Fig. 2E), RAB8a (New Fig. 2G), EXOC1 (New Fig. 4C), and EXOC6 (New Fig. 4D) with post-Golgi carriers.

6. Readout for plasma membrane arrival. They claim of glycosylation of LAMP is used as an indicator of arrival at the PM may not be correct. It is possible that the glycosylation seen happens at the Golgi. To support their point they should show that BFA or 20 degree TGN blocks the glycosylation. If glycosylation still occurs after BFA then the readout would not follow a Golgi to plasma membrane step. If so their assay or interpretation would need to be adjusted.

As discussed above, the glycosylation of LAMP1 is not used as a readout of plasma membrane arrival, and we completely agree with the Reviewer that this would be an inappropriate measure of plasma membrane arrival of cargo since glycosylation of LAMP1 occurs in the Golgi.

Cell surface arrival of GFP-LAMP1 Δ YQTI is detected after RUSH by incubation with a GFP-nanobody that is fused to an mCherry. Thus, our evidence of plasma membrane arrival is drawn from highly quantitative FACS data where each biological experimental repeat is ratiometrically (cell surface/total) calculated from 30,000 individual cells with custom in-house python analysis pipelines to perform individual cell ratios. Each experiment is properly controlled (with a minus biotin baseline condition per sample) and repeated in a minimum biological triplicate. Additionally, we can directly observe the LAMP1 Δ YQTI RUSH cargo trafficking from the ER to the Golgi apparatus (where it is glycosylated), in post-Golgi carriers and to the plasma membrane, where we can directly observe fusion via TIRF microscopy.

We take responsibility for and apologise for this confusion, and to avoid the readership also drawing this incorrect conclusion, we have revised the text in this section and the figure legend and moved the glycosylation, which solely demonstrates the passing of the cargo through the Golgi and proper processing, to the supplement. We thank the Reviewer for this suggestion which we feel helps to clarify the interpretation of this data.

The updated section reads as the following (page 3, lines 98-108):

“To quantitatively study this secretory route we have developed a flow cytometry-based quantitative cell-surface protein delivery assay (Fig.1F). Cells stably expressing LAMP1 Δ YQTI-RUSH were incubated with biotin across a time series of 1-hour, the previously established time to achieve a steady-state level¹. By labelling intact cells with anti-GFP nanobody fused to a fluorescent mCherry⁴⁸ and quantitatively measuring both mCherry and GFP levels via flow cytometry, we can detect and quantify LAMP1 Δ -RUSH delivery to the plasma membrane. The ratio of the cell surface (mCherry) to total cargo available (GFP) then allows for per-cell quantification of protein cargo arrival at the plasma membrane (Fig.1G). In line with previous data, the cargo reaches a steady state ~1 hour after release from the ER with biotin. At 35 minutes, the assay has a high signal-to-noise but is also sensitive to kinetic changes in trafficking, and this time point was selected for future assays. In summary, the use of the RUSH system with a LAMP1 Δ YQTI cargo allows for observation of post-Golgi tubular carriers and their fusion with the plasma membrane with appropriate trafficking kinetics and a quantitative read-out.”

7. Fig. 5. The key data is from "C", but what is analyzed, the change in glycosylation? See above but how do they know the block was earlier than fusion? Also in Fig. 5D the authors should quantitate the levels of cargo inside the cells in +/- EXOC3 to see how this compares with the glycosylation.

We apologise for the confusion in our manuscript here. This is not the change in the glycosylation suggested by the Reviewer but an unbiased FACS analysis of loss of the soluble cargo from the same cell lines used in panel D of this figure. Each data point represents a minimum of 30k cells and is thus an unbiased and experimentally novel quantification of the data in the figure supporting the conclusion that less cargo is secreted from all tested cargoes. Panel D demonstrates that the cargo accumulates in post-Golgi carriers at the cell tips, with an identical phenotype as observed for LAMP1 upon transient exocyst KO. We have clarified the text and the figure legend to avoid this interpretation; we agree the previous figure legend was ambiguous, which we have accordingly updated, and we apologise for the lack of clarity here in the original submission.

The updated section reads as the following (page 9, lines 191-196):

“Intracellular cargo abundance was detected by flow cytometry 12 hours after the addition of biotin. Cargo abundance was compared to a non-biotin condition to calculate the amount of cargo secreted by the cells. By incubating the cells with HALO ligand prior to the addition of biotin, we can remove the effect of nascently synthesised cargo in the assay. All cargoes demonstrated a loss of cargo after RUSH, indicating that all proteins were secreted from the cell, which was significantly decreased with the deletion of exocyst (Fig. 5C).”

And the legend now reads as (page 10, figure legend):

“Flow cytometry-based analysis of protein secretion 12 hours after biotin addition of indicated RUSH cargoes in wt and EXOC3 KO cells. Each condition is normalised to 0 h control resulting in a percent of cargo secreted per sample and condition. Minimum of 30,000 cells per biological repeat, 3 repeats per sample (blue dots). Statistical comparison = *t*-test.”

8. *Nomenclature/Approach.* Exocyst is being used as a term when mainly EXOC3 is examined. They assume that all subunits have similar roles, but several papers (uncited) show that Sec10 had a differential role in MDCK cells. Really other subunits should be examined and the cells should be rescued, ideally in an acute manner using perhaps release of a mitochondrial targeted exocyst subunit. This is a heavy lift, but the authors are not circumspect in the limitations of their approach and in context relative to other prior work.

We thank the Reviewer for these suggestions. To address this in the revision we have new data demonstrating that loss of any of the subunits of the canonical octamer causes a loss in cell surface arrival of post-Golgi cargo (New Fig. 3A), which supports the idea that abrogating any of the eight subunits is suitable to study its role in constitutive secretion. In addition, we have now repeated the carrier-IP experiment with all subunits of the octamer and demonstrate that they are all recruited to the post-Golgi carriers (New Fig. 4F). We thus conclude that indeed the whole complex is recruited rather than one of the described sub-complexes. As Reviewer two suggested, we have now used EXOC3 as a proxy for the complex for imaging experiments and endogenous knock-outs (New Fig. 3A,B, 4A,B,F, 5C,D, 5A,B,D,F,G,H). We have also added the citations suggested by the Reviewer and a section to the discussion about the existence of sub-complexes.

The new section in the discussion reads as the following (page 13, lines 295-305):

“There are a plethora of associated proteins that potentially allow for differential recruitment of exocyst to various carriers. These include the RALs⁵⁶, ARF6⁵⁶, phospholipids⁵⁶ (Fig. 4G,H), CDC42⁸⁸, RAB10⁸⁹, RAB11⁹⁰ and Rab8A⁹¹. In addition, EXOC3 has three homologues *EXOC3L1*, *EXOC3L2* and *EXOC3L4*; and EXOC6 has *EXOC6B*. For example, in this study, EXOC6 acts redundantly with EXOC6B for constitutive cargo delivery, however, absence of EXOC6B alone is sufficient to cause a skeletal disorder in humans^{73,74}. In addition, exocyst subunits have a differential tissue expression in various metazoa^{92,93} and there are a number of functional sub-complexes ascribed to specific cellular functions, including the existence of sub-complex 1 (EXOC1-4) and 2 (EXOC5-8)²⁸, as well as specialised sub-complexes including an EXOC8-dependent sub-complex³⁸, an EXOC2-EXOC8 containing sub-complex^{94,95}, and a specialised role for EXOC7⁹⁶ and EXOC5³¹. Which combination of these sub-complexes, homologues or associated proteins provide specificity, redundancy or regulation of exocyst is not fully understood, but could potentially explain the widespread function of the complex with discrete specificities.”

9. *Functional effect? Which stage of secretion is potentially blocked in the KO? Tethering? Fusion? This would seem to be an important element to characterize.*

Major work has been undertaken on the exact molecular timings of recruitment of exocyst prior to cell surface fusion by the Macara lab (eg. PMID: 30510181), the Murray lab (eg. PMID: 35609603) and others. Due to the nature of the defect upon loss of exocyst we can conclude from our data that exocyst loss causes a defect after vesicle budding and before fusion to the plasma membrane. The colocalisations are before fusion to the cell surface, and as they are on the cell periphery, we conclude that the complex is recruited just prior to fusion, in line with previous evidence from the Macara lab (PMID: 30510181). We have introduced these points in the revised discussion of the manuscript, and we thank the Reviewer for the suggestion. We wish to stress that the focus of the manuscript is on the novel observation that exocyst is essential for soluble protein sorting and not on the molecular architecture and timing of exocyst assembly, although we note our findings are consistent with the literature on the assembly of exocyst onto other carriers.

The discussion now has this paragraph (page 12, lines 266-269):

“Major work has been undertaken on the exact molecular timings of recruitment of exocyst prior to cell surface fusion which show an exquisite order of complex assembly on carriers, upstream of SNARE complex activity and prior to cell surface fusion^{28,70}. Our work is consistent with the observation that the exocyst complex is recruited to secretory carriers near the plasma membrane, prior to their fusion with the plasma membrane²⁸.”

Minor:

• *Video 1E looks encouraging, but why is the movie for only 7 seconds shown? Why not show the full movie (e.g. several minutes?) as done elsewhere?*

We agree with the Reviewer here; the original movie is significantly longer than this. However, due to the high frame rate, large frame size and high image quality, we are restricted by the JBC video upload limit. If the editor wishes us to upload a longer version, we are more than happy to oblige.

• *NPY. This and other soluble probes should be used to see if they use the exocyst? If they do not, how do they rationalize it especially as this cargo was reported to use ELKS as a tether (?)*

NPY is a regulated secreted cargo, not a constitutively trafficked cargo, and thus takes a different secretory pathway in the cell (PMID: 10428060, 8396008), and we, therefore, consider it inappropriate for our study. The regulated secretory pathway is a specialised secretory route from certain cell types, including beta cells and neurons, thus requiring different machinery for constitutive secretion (PMIDs: 32317144, 2994224, 6279313). We wish to stress that we are not suggesting that ELKS plays no role in any secretory pathway. Perhaps it is essential for regulated secretion, as the Reviewer and other literature suggest. We study many constitutively secreted soluble probes in the manuscript, including NUCB1, CAB45, COLX, PAUF, untagged HALO, endogenous adiponectin, endogenous leptin, and endogenous IGGs, as well as performing unbiased whole proteome secretomics.

Overall there are some intriguing findings and if it was true that most soluble cargo require the exocyst to secrete than this would be important. However, to make this claim really requires a high burden of proof - which for the reasons noted above this manuscript has not hit the level of rigor needed for JCB.

We thank the Reviewer for this thorough and thoughtful review. We hope the Reviewer agrees that our revisions and clarifications have met the standard of evidence demanded, and we have demonstrated that, indeed, most soluble constitutively trafficked cargoes require exocyst for secretion.

Reviewer 2

In this manuscript, the authors use a variety of experimental approaches to study the sorting route from the Golgi apparatus to the plasma membrane. Suppression of core exocyst subunits reduced carrier fusion and caused cargo accumulation in the post-Golgi carriers. Overall, the authors make a case that the exocyst complex is a critical component of the mammalian constitutive secretory pathway. The findings are interesting. The work would benefit by addressing several issues.

We thank the Reviewer for their considered comments and their belief that our findings are of interest.

General comments:

1. A brief literature summary of some noncanonical functions of the exocyst such as innate immune function and autophagy would be relevant in the discussion. Perhaps the constitutive and homeostatic nature of these secretory vesicles plays a specific role in cell biologically relevant contexts. Discussing this in a paragraph would improve the manuscript from a cell biological context.

This is an excellent suggestion. We have added a paragraph to the discussion that covers this. We thank the Reviewer for this suggestion.

The discussion now reads as the following:

“Exocyst was initially associated with basolateral vesicle trafficking to cell-cell contacts in polarised cells^{30-34,77-80}, and has since been implicated in a variety of processes including cytokinesis⁴¹⁻⁴³, cell migration and tumour invasion⁸¹⁻⁸⁷, autophagy³⁸ and lysosome secretion³⁹, innate immune response following viral infection⁴⁰ and primary ciliogenesis³⁵⁻³⁷, though in most cases, exocyst’s role is directly linked to its exocytic function²⁶.”

Additionally we have updated this section of the discussion, which we think addresses this point (page 13, lines 295-305):

“There are a plethora of associated proteins that potentially allow for differential recruitment of exocyst to various carriers. These include the RALs⁵⁶, ARF6⁵⁶, phospholipids⁵⁶ (Fig. 4G,H), CDC42⁸⁸, RAB10⁸⁹, RAB11⁹⁰ and Rab8A⁹¹. In addition, EXOC3 has three homologues EXOC3L1, EXOC3L2 and EXOC3L4; and EXOC6 has EXOC6B. For example, in this study, EXOC6 acts redundantly with EXOC6B for constitutive cargo delivery, however, absence of EXOC6B alone is sufficient to cause a skeletal disorder in humans^{73,74}. In addition, exocyst subunits have a differential tissue expression in various metazoa^{92,93} and there are a number of functional sub-complexes ascribed to specific cellular functions, including the existence of sub-complex 1 (EXOC1-4) and 2 (EXOC5-8)²⁸, as well as specialised sub-complexes including an EXOC8-dependent sub-complex³⁸, an EXOC2-EXOC8 containing sub-complex^{94,95}, and a specialised role for EXOC7⁹⁶ and EXOC5³¹.

Which combination of these sub-complexes, homologues or associated proteins provide specificity, redundancy or regulation of exocyst is not fully understood, but could potentially explain the widespread function of the complex with discrete specificities.”

Specific comments:

1. “After synthesis in the ER, LAMP1 traffics via the Golgi apparatus to the plasma membrane where it is endocytosed, in clathrin-coated vesicles, to be delivered to the endolysosomal system and finally to the lysosome”

The authors focus on the claim that defective LAMP1 accumulates on the plasma membrane. However, direct accumulation of LAMP1 on early/sorting endosome has previously been reported. Perhaps the defective LAMP1 has only a membrane directed rout available for these structures. The authors should nevertheless clarify this.

Although the Reviewer is correct that LAMP1-full length has been suggested to be via the endolysosomal system as well as the indirect route via the PM, we clearly observe LAMP1-deltaYQTI trafficking directly to the plasma membrane from the Golgi apparatus, in line with previously published findings (PMID: 28978644), we address this in figure 1 of the manuscript demonstrating that a statistically significant amount of LAMP1-deltaYQTI traffics directly to the plasma membrane. We have clarified this point in the manuscript as suggested by the reviewer.

The new paragraph in the discussion reads as the following (page 12, lines 245-251):

“We chose LAMP1ΔYQTI as a probe as it has been experimentally demonstrated to traffic directly to the cell surface via the biosynthetic secretory pathway¹, and thus can act as an orthologous validated marker of this pathway, as demonstrated in Figures 1 and 2. Additionally, LAMP1 can tolerate a tag on the N and C termini without affecting the trafficking or kinetics¹ (Fig. S4). Some studies suggest that LAMP1 traffics via the endolysosomal system to the lysosome⁶² (the so-called ‘direct pathway’), and we cannot rule out a non-detectable subset of the RUSH cargo taking this pathway. Nevertheless, with the LAMP1ΔYQTI-RUSH we observe a significant amount of trafficking directly to the cell surface (Fig. 1) in line with other studies¹.”

2. *EXOC1-HALO experiment: The authors do not mention why EXOC1 (Sec3) was selected for in this analysis. Generally, in mammalian cells EXOC3 (Sec6) and EXOC4 (Sec8) (Exocyst is also called Sec6/8 complex), are used as a proxy for the holocomplex, since several subcomplexes of the exocyst has been reported. Specifically, secretory vesicle fusion by the exocyst requires two distinct complexes coming together and only EXOC3 and EXOC4 have been shown to be part of both of them, where EXOC3 is not. The authors need to clarify.*

We thank the Reviewer for this comment. To address this, taking into account the advice of the Reviewer, we have repeated many of the experiments using EXOC3 as a proxy (New Fig. 3A,B, 4A,B,F, 5C,D, 6A,B,D,F,G,H). The data in the original manuscript has been, for the most part, retained in the revised manuscript; however, we now use EXOC3 as a common thread throughout the manuscript for specific experiments. This includes the TIRF colocalisation, which benefited from studying a KO/recovery condition to prevent unlabeled endogenous exocyst from blocking the localisation (New Fig. 4A). In addition, the adipocyte leptin secretion data in figure 6 has been repeated with *Exoc3* knockdown (Fig. 6D,F). In addition, to prevent us from misinterpreting a sub-complex based on EXOC3, we have repeated the key experiments with all major exocyst subunits. The flow cytometry-based cell surface assay has been repeated with EXOC1, EXOC2, EXOC3, EXOC4, EXOC5, EXOC6/B, EXOC7 and EXOC8 (New Fig. 3A), as has our carrier IP approach (New Fig. 4F). We feel these experiments have resulted in a substantive improvement in the manuscript’s readability and cohesiveness, and we thank the Reviewer for suggesting these experiments.

3. *Notably, EXOC4/Sec8-RFP based TIRF microscopy tools have been used previously and well established in the field. EXOC4 modifications were shown to not affect holocomplex formation as well, which is not the case for EXOC1. The authors should justify the selection of EXOC1 in this regard. Does heterologous EXOC1 localize in the holocomplex? The authors should show this.*

We have used EXOC1 as a proxy for the complex based on its ability to localise to our carrier and the fact that KO causes a similar phenotype to the rest of the complex in our system, As discussed above, to generate a more cohesive study and complement the existing literature, we have repeated this experiment with EXOC3. We have demonstrated through our recovery experiments that EXOC3-HALO is functional (New Fig. 3A), and by performing these experiments under KO recovery conditions with low expression, we could reproducibly observe statistically significant recruitment of EXOC3 to the carriers (Fig. 4A,B).

Also why did the authors switch to EXOC6 (Sec15) for the subsequent assay? Consistency in prioritizing the subunits for the assay is crucial since their participation in subcomplexes can vary. The authors should explain and preferably perform experiments solely with canonical EXOC3/Sec6 and EXOC4/Sec8 for both of these assays.

We agree with the Reviewer that it is more desirable to have consistent use of subunits throughout our manuscript. As stated above, we have now added additional data using EXOC3 as a proxy for the complex throughout our paper so that all experiments now use a conical and consistent subunit. Additionally, for key experiments (the recruitment to the carriers and the effect on cell surface delivery), we have added data for every subunit of the complex. We again thank the Reviewer for these comments, as we think this has substantially improved the clarity of the manuscript.

4. Carrier IP: Authors should also show EXOC2 (Sec5) and EXOC8 (Exo84). These are two of the most important members of the exocyst subcomplexes reported.

We have repeated our carrier-IP experiments and probed for all exocyst subunits as advised by the Reviewer and revised the manuscript accordingly (Fig4F). We observe that all exocyst subunits are recruited to the carriers. We thank the Reviewer for this suggestion, as these data have provided strong evidence for the recruitment of the complete exocyst complex to post-Golgi secretory carriers.

5. The authors should show validation for the RALA and RALB KO validation, since RALGTPases, (RALA and RALB) are the major, if not, the prime regulators of the exocyst in mammalian cells validated by prior literature. PIP5K data is interesting, but lacks depth as a stimulatory node upstream. PIP5K perhaps is mediating docking of the exocyst complex, not formation of the holocomplex. If otherwise, the authors should delineate how PIP5K can drive exocyst holocomplex assembly at the tethering site.

We thank the Reviewer for bringing this up. We have indeed tested our KO of the Rals using qPCR, and we achieved a 95% KO in each case; however, we cannot exclude that the remaining protein is functional or if there are additional redundant regulators or specific regulators for this process.

Please see our response to Reviewer 1 above: We agree with this point that the loss of both RALs is not as severe a phenotype as the loss of other essential components in this study. This could be due to undiscovered redundancy, the ability of the remaining protein pool, despite 95%+ loss of RNA levels, to compensate, or another mechanism. As our manuscript does not concern the role of the Rals, we have removed this from the updated submission; we feel this helps with the clarity of the manuscript for the readership and thank the Reviewer for the suggestion.

In addition to the response above, we also note that the enzymatic activity of PIP5K has recently been demonstrated to be sufficient for exocyst binding to the membrane (PMID:35609603). We agree that PI5K is most likely important for complex recruitment to carriers. We note that our manuscript is specifically focused on demonstrating that the exocyst has a role in the fundamental process of constitutive secretion, a finding we agree with Reviewer 3 fills an “important gap in the knowledge of the function of the exocyst and brings solid evidence of its involvement in direct secretory trafficking from the Golgi to the plasma membrane”.

6. The exocyst's functional role as a major machinery for secretion is well understood. This function has been reported to be non-specific and constitutive. The novelty of last two figures in this regard is unclear.

The Reviewer is correct that exocyst has been implicated in constitutive secretion in yeast. In mammalian cells, however, exocyst has not been previously demonstrated to play a role in constitutive soluble protein secretion. Secretion in mammalian cells is significantly more complicated as there are at least 3 independent secretory pathways (constitutive, regulated dense-core and regulated synaptic), and cell type specialities resulting in unique membrane trafficking pathways (for example, in the case of polarised sorting).

The protein associated with constitutive soluble protein secretion in mammalian cells is a long coil-coiled protein named ELKS (PMID: 31142554).

We agree that exocyst is well characterised in endosomal recycling to the cell surface in mammalian cells (i.e. from the endosome to the cell surface). However, this is a fundamentally different pathway with different carriers and lipid components than post-Golgi biosynthetic protein secretion. As we demonstrate in the manuscript, the cargos that take this pathway include collagen (which accounts for 30% of protein mass by dry weight in mammals), antibodies, and hormones (e.g., leptin). We thus consider the last two figures the

major novel point of the manuscript as they demonstrate the role of exocyst in endogenous soluble protein secretion in mammalian cells – a novel finding.

Reviewer 3

In this manuscript Pereira et al. use a combination of trafficking assays, imaging and transient CRISPR-Cas9 knock out to test the involvement of the exocyst in constitutive secretion. Using a LAMP1 mutant lacking the endocytic trafficking motif as a reporter for quantitative RUSH assays they show that secretory carriers traffic on microtubule tracks and are positive for the small GTPase Rab6 and Rab8 and ARHGEF10. KO of Rab6 causes accumulation of the reporter in the Golgi. Further employing their assays, they show that the exocyst subunit EXOC1 localized to fusion hot spots together with the secretory carriers and KO of two exocyst subunits EXOC3 and 5 causes accumulation of the LAMP1 reporter in the cell periphery. Interesting when KO the other plasma membrane tether ELKS, secretion of the reporter is not affected. Taking advantage of a cytoplasmic tag on the reporter they are able to purify the transport carriers and show enrichment of exocyst subunits and the small GTPase Rab6. KO of PIP5Ks and RalA/B affected secretion of the reporter. The authors then nicely show that EXOC3 KO disrupt secretion of various known secretory markers. Employing mass spectrometry to study the secretome of EXOC3 KO cells they show that many soluble cargoes cannot reach the cell surface. KO of EXOC3 in adipocytes and lymphocytes blocks the secretion of physiological cargoes, strengthening the message of the paper.

Overall, this manuscript fills an important gap in the knowledge of the function of the exocyst and brings solid evidence of its involvement in direct secretory trafficking from the Golgi to the plasma membrane using solid kinetic assays and also providing specific disruption of the endogenous secretome. The paper is very nicely written and easy to read, experiments are well executed and presented. Therefore, I would recommend acceptance of the manuscript with the following minor revisions:

We thank the Reviewer for their summary of our manuscript and positive comments.

- Figure 2: color coding in the crops is a bit confusing. I understand that the authors want to use the colors of the labels to frame the crop so shouldn't the reporter crop green? It is cyan at the moment.

This was an oversight, and we thank the Reviewer for noticing. We have corrected this error.

- Figure 2: Arrows to highlight the co-localizing structures would help.

We agree and have modified the figure accordingly.

- Supplementary Figure 1: hard to see the co-localization with this LUT and without single color images/crops.

We have amended this according to the Reviewer's suggestion.

- Interesting that ELKS localizes to the same hot spots as EXO and the carries but KO of ELKS does not affect secretion of the reporters at all? Is this worth mentioning in the discussion?

We agree this is interesting. We have added a comment on this in the discussion. We note that although ELKS does not appear essential, a double KO of EXOC1 and ELKS demonstrates an additional, significant decrease in cell surface delivery, suggesting that ELKS may be involved in this process, despite being non-essential according to our findings.

The discussion now includes this (page 12, lines 270-282):

"ELKS has previously been identified as the molecular tether for secretory carriers²². Although we observed ELKS localising to 'hot-spots' on the plasma membrane, we did not see a phenotype of ELKS-KO on cell surface delivery of LAMP1Δ-RUSH (Fig. S2). This does not rule out the role of ELKS in this process and it is likely that for certain cargoes or cell types ELKS has an essential role. Accordingly, we observe an increase in the phenotype when combining ELKS and exocyst knock-out (Fig. S2F) indicating a potential functional redundancy, and notably see both ELKS and exocyst on the same 'hot-spots' where cell surface fusion occurs. Indeed, in neurons, studies demonstrate that ELKS acts as a redundant scaffold protein at the active zone site and that when this structure is disrupted, synaptic vesicle fusion is impaired⁷¹. Additionally, one of the cargoes that ELKS has been shown to have a role in the secretion of is NPY, a soluble secreted cargo that takes the regulated secretory pathway in specialised cell types²². Besides a structural role, ELKS has been further

shown to capture RAB6 positive cargoes in golgin-like manner contributing to the establishment of a ready to fuse pool of synaptic vesicles at the active zone²⁵. To fully understand the specific balance of roles between exocyst and ELKS will require further studies of other cargoes in specific cell types.”

- *Figure 3A: It would be useful to have a crop to better show the co-localization. Hard to see with a low quality overview image.*

We thank the Reviewer for this comment. Figure 3A is now in supplement and we have an updated figure with EXOC3, where we have arrows indicating points of colocalisation, and a quantification of the localisation. We performed this new imaging in a EXOC3 KO background, which allowed us to produce clearer colocalisation data.

- *Figure 4A: Aren't figure 3A and 4A showing the same thing with two different exocyst subunits? Again crops and arrows would be useful.*

We agree with this comment. We have now modified the manuscript moving this data to the supplement, and adding new data on EXOC3 in figure 4 and added new statistics on the previous data into this figure as well.

- *Figure 4D-E While the effect of PIP5K is obvious the effect of the double RalA/B KO is only modest. The authors should carefully rephrase their conclusions.*

As stated to the other Reviewers, we have elected to remove the Ral data from the manuscript (original figures 4D-E). These data are not important for our fundamental conclusions, and as stated above, there are many possible reasons why the phenotype is modest. We thank the Reviewer for suggesting this.

- *Line 172: I cannot find any supplementary figure 4.*

We apologise; this was a typographical error. The correct figure reference is figure S3 in the original submission. This has been fixed in the revised version.

- *Line 172: "We then performed RUSH assays and used an unbiased FACS assay to quantitatively monitor the loss of these soluble cargoes". Does it mean that quantification in figure 5C was carried out in a different way as explained in 1G? The authors could explain in the figure legend? Are all RUSH assays quantified using FACS as explained in the methods?*

Yes, this is a different assay from that explained in the original Fig 1G. We apologise for the confusion here. In this case, we quantified the amount of the soluble HALO-tagged proteins *lost* from the cells using FACS analysis as a measure of secretion. We have updated the text and figure legend to communicate this better.

The updated section reads as the following (page 9, lines 191-196):

“Intracellular cargo abundance was detected by flow cytometry 12 hours after the addition of biotin. Cargo abundance was compared to a non-biotin condition to calculate the amount of cargo secreted by the cells. By incubating the cells with HALO ligand prior to the addition of biotin, we can remove the effect of nascently synthesised cargo in the assay. All cargoes demonstrated a loss of cargo after RUSH, indicating that all proteins were secreted from the cell, which was significantly decreased with the deletion of exocyst (Fig. 5C).”

And the legend now reads as (page 10, figure legend):

“Flow cytometry-based analysis of protein secretion 12 hours after biotin addition of indicated RUSH cargoes in wt and EXOC3 KO cells. Each condition is normalised to 0 h control resulting in a percent of cargo secreted per sample and condition. Minimum of 30,000 cells per biological repeat, 3 repeats per sample (blue dots). Statistical comparison = *t*-test.”

- *In general: bars in graphs are SD? SE?*

Error bars are SD; we have now included this information in figure legends.

We thank all the Reviewers for their thoughtful and useful comments. We believe the resubmitted manuscript has been strengthened thanks to this discourse. We look forward to hearing from you in due time regarding our submission and to respond to any further questions and comments you may have.

Sincerely,

David Gershlick

--
Sir Henry Dale Fellow,
University of Cambridge
Cambridge Institute for Medical Research,
Cambridge Biomedical Campus,
Hills Road,
Cambridge
CB2 0XY, UK
www.gershlicklab.com

December 14, 2022

RE: JCB Manuscript #202205137R

Dr. David C. Gershlick
University of Cambridge
Cambridge Biomedical Campus
Cambridge CB2 0XY
United Kingdom

Dear Dr. Gershlick:

Thank you for submitting your revised manuscript entitled "The exocyst complex is an essential component of the mammalian constitutive secretory pathway". We would be happy to publish your paper in JCB pending final revisions necessary to meet our formatting guidelines (see details below).

A. MANUSCRIPT ORGANIZATION AND FORMATTING:

Full guidelines are available on our Instructions for Authors page, <http://jcb.rupress.org/submission-guidelines#revised>. Submission of a paper that does not conform to JCB guidelines will delay the acceptance of your manuscript.

1) Text limits: Character count for Articles is < 40,000, not including spaces. Count includes abstract, introduction, results, discussion, and acknowledgments. Count does not include title page, figure legends, materials and methods, references, tables, or supplemental legends.

2) Figures limits: Articles may have up to 10 main figures and 5 supplemental figures/tables.

3) Figure formatting: Scale bars must be present on all microscopy images, including inset magnifications. Molecular weight or nucleic acid size markers must be included on all gel electrophoresis.

** Please add scale bars to inset images as some of these show significantly smaller fields than the parent image.

** Please indicate molecular weight markers in Supplemental Figures 1 and 2C.

4) Statistical analysis: Error bars on graphic representations of numerical data must be clearly described in the figure legend. The number of independent data points (n) represented in a graph must be indicated in the legend. Statistical methods should be explained in full in the materials and methods. For figures presenting pooled data the statistical measure should be defined in the figure legends. Please also be sure to indicate the statistical tests used in each of your experiments (either in the figure legend itself or in a separate methods section) as well as the parameters of the test (for example, if you ran a t-test, please indicate if it was one- or two-sided, etc.). Also, if you used parametric tests, please indicate if the data distribution was tested for normality (and if so, how). If not, you must state something to the effect that "Data distribution was assumed to be normal but this was not formally tested."

** Please indicate whether t-tests were one- or two-tailed in all figure legends.

** Please indicate the statistical test used in Figures 2H, 3A, 4G, 6E, 6F, 6H, and Supplemental Figures 2B, 2D, 2F, 3D, 3E,

** Please indicate n for Figures 4G, 6H, and Supplemental Figures 2B, 2D, 2F, 3D, 3E,

5) Abstract and title: The abstract should be no longer than 160 words and should communicate the significance of the paper for a general audience. The title should be less than 100 characters including spaces. Make the title concise but accessible to a general readership.

6) Materials and methods: Should be comprehensive and not simply reference a previous publication for details on how an experiment was performed. Please provide full descriptions in the text for readers who may not have access to referenced manuscripts.

7) Please be sure to provide the sequences for all of your primers/oligos and RNAi constructs in the materials and methods. You must also indicate in the methods the source, species, and catalog numbers (where appropriate) for all of your antibodies. Please also indicate the acquisition and quantification methods for immunoblotting/western blots.

8) Microscope image acquisition: The following information must be provided about the acquisition and processing of images:

a. Make and model of microscope

b. Type, magnification, and numerical aperture of the objective lenses

- c. Temperature
- d. Imaging medium
- e. Fluorochromes
- f. Camera make and model
- g. Acquisition software
- h. Any software used for image processing subsequent to data acquisition. Please include details and types of operations involved (e.g., type of deconvolution, 3D reconstitutions, surface or volume rendering, gamma adjustments, etc.).

10) Supplemental materials: There are strict limits on the allowable amount of supplemental data. Articles may have up to 5 supplemental figures. Please also note that tables, like figures, should be provided as individual, editable files. A summary of all supplemental material should appear at the end of the Materials and methods section.

13) ORCID IDs: ORCID IDs are unique identifiers allowing researchers to create a record of their various scholarly contributions in a single place. At resubmission of your final files, please consider providing an ORCID ID for as many contributing authors as possible.

Please note that JCB now requires authors to submit Source Data used to generate figures containing gels and Western blots with all revised manuscripts. This Source Data consists of fully uncropped and unprocessed images for each gel/blot displayed in the main and supplemental figures. Since your paper includes cropped gel and/or blot images, please be sure to provide one Source Data file for each figure that contains gels and/or blots along with your revised manuscript files. File names for Source Data figures should be alphanumeric without any spaces or special characters (i.e., SourceDataF#, where F# refers to the associated main figure number or SourceDataFS# for those associated with Supplementary figures). The lanes of the gels/blots should be labeled as they are in the associated figure, the place where cropping was applied should be marked (with a box), and molecular weight/size standards should be labeled wherever possible.

WHEN APPROPRIATE: The source code for all custom computational methods published in JCB must be made freely available as supplemental material hosted at www.jcb.org. Please contact the JCB Editorial Office to find out how to submit your custom macros, code for custom algorithms, etc. Generally, these are provided as raw code in a .txt file or as other file types in a .zip file. Please also include a one-sentence summary of each file in the Online Supplemental Material paragraph of your manuscript.

B. FINAL FILES:

-- Cover images: If you have any striking images related to this story, we would be happy to consider them for inclusion on the journal cover. Submitted images may also be chosen for highlighting on the journal table of contents or JCB homepage carousel.

Images should be uploaded as TIFF or EPS files and must be at least 300 dpi resolution.

****It is JCB policy that if requested, original data images must be made available to the editors. Failure to provide original images upon request will result in unavoidable delays in publication. Please ensure that you have access to all original data images prior to final submission.****

****The license to publish form must be signed before your manuscript can be sent to production. A link to the electronic license to publish form will be sent to the corresponding author only. Please take a moment to check your funder requirements before choosing the appropriate license.****

Thank you for this interesting contribution, we look forward to publishing your paper in Journal of Cell Biology.

Sincerely,

Ian Macara, Ph.D.
Editor
The Journal of Cell Biology

Tim Fessenden
Scientific Editor
Journal of Cell Biology

Reviewer #1 (Comments to the Authors (Required)):

The manuscript is considerably improved in the revision with the new experiments, quantification and rewording and all my critiques have been adequately addressed.

Reviewer #2 (Comments to the Authors (Required)):

The authors have submitted a revised manuscript that is improved. They have addressed my comments.